# Structural conservation of insulin/IGF signalling axis at the insulin receptors level in *Drosophila* and humans

Cristina M. Viola[1,4], Orsolya Frittmann[1,5], Huw T. Jenkins [1], Talha Shafi[1], Pierre De Meyts [2,3] & Andrzej M. Brzozowski [1] ✉

The insulin-related hormones regulate key life processes in Metazoa, from metabolism to growth, lifespan and aging, through an evolutionarily conserved insulin signalling axis (IIS). In humans the IIS axis is controlled by insulin, two insulin-like growth factors, two isoforms of the insulin receptor (hIR-A and -B), and its homologous IGF-1R. In *Drosophila*, this signalling engages seven insulin-like hormones (DILP1-7) and a single receptor (dmIR). This report describes the cryoEM structure of the dmIR ectodomain:DILP5 complex, revealing high structural homology between dmIR and hIR. The excess of DILP5 yields dmIR complex in an asymmetric '*T*' conformation, similar to that observed in some complexes of human IRs. However, dmIR binds three DILP5 molecules in a distinct arrangement, showing also dmIR-specific features. This work adds structural support to evolutionary conservation of the IIS axis at the IR level, and also underpins a better understanding of an important model organism.

The insulin/insulin-like growth factor signalling axis (IIS) is an evolutionarily ancient, highly conserved, endocrine and paracrine signal transduction network in Metazoa[1,2]. IIS regulates a wide range of life processes such as growth, metabolism, development, aging and lifespan, reproduction, and cell growth, differentiation and migration.

In many vertebrates including humans, insulin is stored in oligomeric crystalline forms in pancreatic β-cells, being rapidly secreted into the circulation in a monomeric form in response to glucose and nutrients[3]. Its homologous human insulin-like growth factors (hIGF-1 and 2) do not aggregate[4], are secreted by several tissues as monomers, occurring in biological fluids in complexes with several IGF binding proteins (IGFBP1-6) that regulate their bioavailability[5]. Ultimately, insulin and IGFs signal through closely related cell-surface receptor tyrosine kinases (RTKs), the insulin and IGF-1 receptors (hIR, hIGF-1R), and through their hybrid dimers[6–9].

In contrast, insulin-like proteins (ILPs) are very diverse in the animal kingdom, ranging from 7 ILPs in *Drosophila* (DILP1-7), to 40 ILPs in *Bombyx mori* and *Caenorhabditis elegans*[6,10–15], while *Drosophila* DILP8 is a paralogue of human relaxin and a ligand of a G-protein-coupled receptor, Lgr3[16]. However, all these hormones share similar motifs of inter-/intra-chain disulfides and human insulin-like organisation, i.e., two A, B chains with the middle C-domain that is either processed from the prohormone, or being retained in certain ILPs, such as human proinsulin and hIGF-1/2 (Fig. 1a, Supplementary Fig. 1). Despite the variety of the ILPs, in humans they act through two isoforms of hIR (hIR-A and -B) and one hIGF-1R receptor (hIGF-1R)[6,7], while *Drosophila melanogaster* (Dm) and *Caenorhabditis elegans* encode a single so-called insulin receptor for many ILPs[10,11,13]. This expands the role of *Drosophila* insulin receptor (dmIR) into many life processes including development, regulation of life span, metabolism, stress resistance, reproduction and growth. It has been proposed that the recently discovered *IR* gene duplication and a decoy (TK-free) dmIR-framework based receptors in some insects play a role in a phenotypic plasticity and caste differentiation in insect taxa[17–19]. Nevertheless,

[1]York Structural Biology Laboratory, Department of Chemistry, University of York, Heslington, York YO10 5DD, UK. [2]Department of Cell Signalling, de Duve Institute, B-1200 Brussels, Belgium. [3]Department of Cell Therapy Research, Novo Nordisk A/S, DK-2670 Maaloev, Denmark. [4]Present address: Department of Biology, University of York, Heslington, York YO10 5DD, UK. [5]Present address: Department of Haematology, University Medical Center Groningen, Hanzeplein 1, 9713 GZ Groningen, Netherlands. ✉e-mail: marek.brzozowski@york.ac.uk

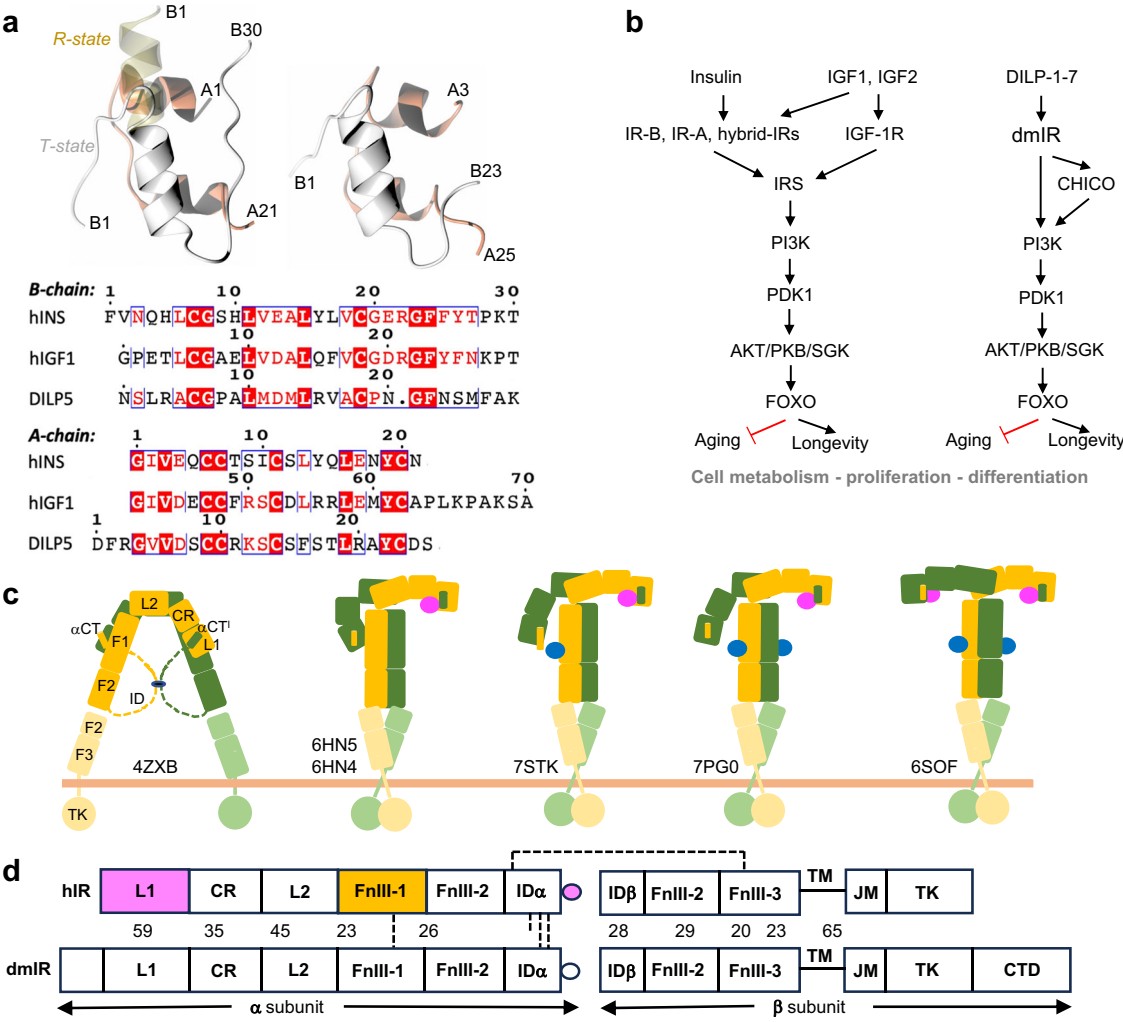

**Fig. 1 | Summary of the conservation of IIS axis in *Drosophila* and humans.**
**a** Structural similarity of human insulin (top left) and DILP5 (right), B-chains in grey, A-chains in coral, pale yellow−insulin's B1-B6 N-terminus in the so-called R-state; bottom – the sequence alignment of human insulin, hIGF-1 (its C-domain is omitted here as not relevant for DILPs) and DILP5. **b** Functional homology and conservation of IIS axis in humans and *Drosophila*, with some representatives of the key and common downstream effector proteins. **c** Schematic representation of the domain organisation, and examples of some key structural conformers, of the hIR in the

*apo* (far left) and its *holo* forms; yellow/green−hIR protomers, insulins in site 1/1' in pink, insulins in site 2/2' in blue; PDB IDs are given for some representative structures. **d** Comparison of the domain organisation of hIR and dmIR αβ protomers (based on the sequence alignment), and their sequence similarity (in %); in magenta −domains involved in insulin binding site 1/1', in yellow−site 2/2'; dashed lines - disulfide bonds; dmIR has only two inter-ID domains -SS- bonds, as it does not have the equivalent of human Cys682 in this region.

close structural and functional homology between mammalian and invertebrates ILPs, and their IRs, is highlighted by the activation of dmIR by human insulin[20,21], the insulin-like bioactivity of DILP5 in mice and *Drosophila*[21], and hIR activation (on the functional and structural level) by the cone snail venom insulin[22].

*Drosophila* is a widely used model for many human pathologies ranging from metabolic disorders to neurodegeneration, as over 75% of human-disease-related genes are conserved in this animal[23,24]. Although the network of the IIS axis is reduced in flies, usually by the presence of only one downstream signalling homologue, i.e., a lower redundancy at the signal amplification levels (Fig. 1b), a direct comparison of the human and Dm IIS axes remains complex[25]. For example, a recent structure of the *Drosophila* insulin binding protein Imp-L2 dispelled the commonly held notion that it is a homologue of human IGFBPs but represents a very different family of insect proteins involved in the regulation of DILPs bioavailability[26].

Early biochemical studies established a largely invariant region of human insulin as a good candidate for the receptor-binding epitope (now referred as to site 1), including A-chain residues GlyA1, GlnA5,

TyrA19, AsnA21, and B-chain residues ValB12, TyrB16, PheB24, PheB25 and TyrB26[27,28]. Subsequently, they were extended to SerA12, Leu A13, GluA17, HisB10, GluB13, GluB17 and IleA10[7,29,30], which form hIR binding site 2. Two binding sites equivalent to insulin's sites 1 and 2 have also been identified on hIGF-1 and 2[31,32], with site 1 extending into their C-domain. The role of these sites was ultimately confirmed in the last decade in over 40 crystal and cryoEM structures of complexes of these hormones (as well as insulin mimetic peptides and aptamers) with their cognate receptors, and their extensive functional studies[22,33–54]; for reviews, see refs. 55,56).

Despite astonishing 3-D insights into the *holo*- and *apo* forms of hIR/hIGF-1R (and mice IR (mIR)), several key questions about their insulin-binding activation and complex allosteric signal transduction still remain. The variety of insulin:IR stoichiometries with a broad spectrum of conformations of the ectodomains (ECD) of these receptors, led to several alternative models of signal transduction through the IRs (Fig. 1c).

The importance of understanding the *Drosophila* IIS axis contrasts with the lack of the structural description of its dmIR, or indeed any

non-mammalian/invertebrate IRs; however, an extensive mutagenesis-based insight into the TK of this receptor is available[57]. This prompted us to undertake the structural characterisation of the dmIR ectodomain (dmIR-ECD) in its free- and liganded-form in the complex with the *Drosophila* DILP5 hormone.

The dmIR primary structure, with ~33% sequence identity in the overlapping regions[58,59], suggests that its domain organisation is similar to human IR-like receptors (Supplementary Fig. 2). dmIR domain topology, similar to hIR, is made of unique $(\alpha\beta)_2$ subunit homodimers, with the same modular, exons-reflecting organisation (Fig. 1d). In hIR (and hIGF-1R) the α-subunits consist of L1 (Leucine-rich repeats domain 1), CR (Cysteine Rich), L2, FnIII-1 (Fibronectin type III-like), FnIII-2 and internally spliced ID (Insert Domain) modules. The β-subunit starts with the remaining part of the spliced ID/FnIII-2 domain, followed by the FnIII-3 domain, the transmembrane helix (TM), juxtamembrane (JM) and TK modules[7]. Both α-subunits are linked by several structurally and functionally important -SS- bonds, while a single -SS-bond connects α−β-subunits[6].

The L1 domain and the crossing-over α−CT′ terminal region of the ID′ domain from the other α-subunit form the main, high affinity, insulin binding site 1 (and 1′) (e.g., L1/α−CT′, L1′/α−CT) (the (′) denotes contribution from the other symmetrical αβ half of the IR) (34). The α−CT segments lie across the L1β2 sheets of the L1 domains, adjusting their *apo*-IR positions to mediate a mostly indirect tethering of insulin onto the L1β2 surface. The so-called insulin low-affinity sites 2/2′ are on surfaces of the FnIII-1/FnIII-1′ domains[55,56]. The dmIR terminates with a unique 60 kDa extension: C-terminal domain (CTD), the N-terminal part of which has 22% sequence identity with human intracellular Insulin Receptor Substrate 1[58,59] (IRS1)(Fig. 1d, Supplementary Fig. 2).

Human and mouse IR structures consistently revealed the spectrum of conformations from the *apo*-IR in an inverted V - Λ-like - form, to a very different - fully four-insulin saturated, almost symmetrical T-shaped form with insulins in each sites 1/1′ and 2/2′ (reviewed in refs. 55,56). Many 'intermediate' hIR/mIR structures have been found between these two extreme IR conformers, with one, two, or three insulins bound[50,56]. These intermediate structures have the α-subunit arms of the receptor in many asymmetrical *Γ* and *T* conformations, from one arm-down/one-up to the different stages of detachment of the down-arm from the FnIII-1−FnIII-3-stem of the receptor (Fig. 1c)[56]. It seems that the ID and ID′ linkers that are inter-connected by three -SS-bonds (two in Dm (Fig. 1d, Supplementary Fig. 2)) correlate the movement of insulin-binding α−CT helices with other parts of the IR into complex allosteric effects[45]. The αβ half of the IR with a fully stretched Γ up-arm and site 1 bound insulin has been referred as the so-called static 'invariant' protomer, as it does not deviate significantly in the *holo*-IR and liganded ECD forms, while the down-arm of the other αβ half of the IR - in different stages of detachment from the stem of the receptor - is described as a dynamic protomer. There is no single explanation of the activation steps of the hIR/mIR yet, but it can be envisaged that binding of insulin to one low-affinity site 2 triggers the sequence of allosteric effects, leading to successive association(s) of the hormone with high-affinity site(s) 1[50,56]. Although only one hIGF-1 hormone is sufficient to activate the hIGF-1R *via* its asymmetric (even in the excess of the ligand) conformer, the general layout and principle of hIGF-1R function parallel the key aspects of hIR activation[44].

We selected the *Drosophila* DILP5 for dmIR-ECD studies as a typical representative and well-characterised fly ILP, having also a high homology to human insulin (~28% sequence identity[21], Supplementary Fig. 1), and with nM-range $K_D$ for the dmIR high-affinity site 1 of 0.76 (±0.16) nM[21] that is within a range of human insulin:hIR interaction (e.g., 0.2−0.7 nM). DILP5 has a very human insulin-like structure (Fig. 1a), and a substantial hIR affinity of $K_D$ ~ 60 nM[21]. Each of the DILP1-7 has a unique tissue/life cycle expression pattern and regulation, with DILP5 being expressed in neurosecretory cells of *Drosophila* brain,

midgut and ovaries, and the ablation of these cells reflects the reduction of insulin signalling in mammals, leading to, for example, a whole-body increase of trehalose in the fly[21,25].

Here, we provide the 3-D description of an invertebrate IR-like receptor by determining the cryoEM structure of the dmIR-ECD in the complex with DILP5. This provides structural evidence of the conservation of the IIS axis in the animal kingdom at the IR level, with dmIR-ECD following the principle and organisation of h(m)IRs and hIGF-1R. However, a handful of dmIR-ECD unique features are shedding light on a possible adaptation of this system to *Drosophila*-specific multi-hormone driven signal transduction through this receptor.

## Results

The dmIR-ECD construct used in this study is a codon optimised sequence derived from Uniprot sequence P09208. Residues 1-263, upstream of a predicted signal sequence (residues 264-290, see Methods), and downstream of the expected 1-1309 ectodomain were not included in the expression construct. A methionine was placed in front of the predicted signal sequence with a C-terminal StrepII-tag and expressed in Sf9 cells using the baculovirus system. The insect cell expressed construct was purified using StrepTrap HP affinity media and size-exclusion chromatography. Pure, native PAGE single band protein was used for further studies. For the dmIR-ECD complex, the DILP5 C4 variant of this hormone (referred here to as DILP5) was used[21] (Supplementary Fig. 1), with AspA1-PheA2-ArgA3 A-chain N-terminal extension in comparison with the so-called DILP5 DB variant; they have been studied with their alternative A-chains due to ambiguity of their in vivo processing sites[21]. DILP5 $K_D$ of 498 nM for this construct was assessed by µITC indicating its low dmIR-ECD affinity. While potentially a feature of the construct used in this study, ITC-based $K_D$ values - much lower affinity than, for example, the $IC_{50}$ competition-based assays derived affinities obtained for IR-like receptor (hIGF-1R) - have been reported[36], with such discrepancy likely resulting from much higher sample concentrations required for ITC. Nevertheless, to assure an effective hormone:receptor engagement the sample of the complex used for the cryoEM studies was prepared with five molar excess of DILP5 to the dmIR-ECD (see Methods).

### General organisation of the dmIR-ECD

The ligand-free form of the dmIR has only been outlined here at a very low resolution as its molecules constituted only a small sub-fraction of the observed particles (Fig. 2a, Supplementary Fig. 3). Nevertheless, it seems that, in overall, the conformations of *apo*-hIRs, its ligand-free ECDs and dmIR-ECD are similar, appearing as Λ-like intertwined $(\alpha\beta)_2$ homodimers. They likely represent the non-signalling forms of the receptor due to a large (~110 Å) separation of its FnIII-3 domains.

The 5:1 molar hormone:receptor ratio used for the preparation of the dmIR-ECD complex resulted in an asymmetric three-DILP5:dmIR-ECD arrangement. There are two adjacent insulins bound to sites 1 and 2′ on the down-arm/dynamic and static protomers, respectively, and one insulin at site 1′ on the arm-up/static protomer (Fig. 2b–d). Such IR conformer has not been seen in any wild-type human insulin:IR complexes; if three-insulins:IR complex was observed there was a clear separation between site 1 and site 2′ bound hormones (e.g., PDB ID: 7PG0)[50].

Despite a large phylogenetic gap between insects and humans[15], the 3-D principles of the overall organisations of h(m)IR/hIGF-1R and dmIR ECDs are very similar (Fig. 2). The structure of the dmIR-ECD $(\alpha\beta)_2$ homodimer – with Asn335 as the first N-terminal residue observed in the maps - follows closely the multidomain order and size of hIR/hIGF-1R ECDs (Fig. 1d, Fig. 2b–d, Supplementary Fig. 2). Therefore, as this report concerns dmIR-ECD complex with a more human insulin-like DILP5, the comparison focuses here on dmIR and hIR for the brevity of the argument.

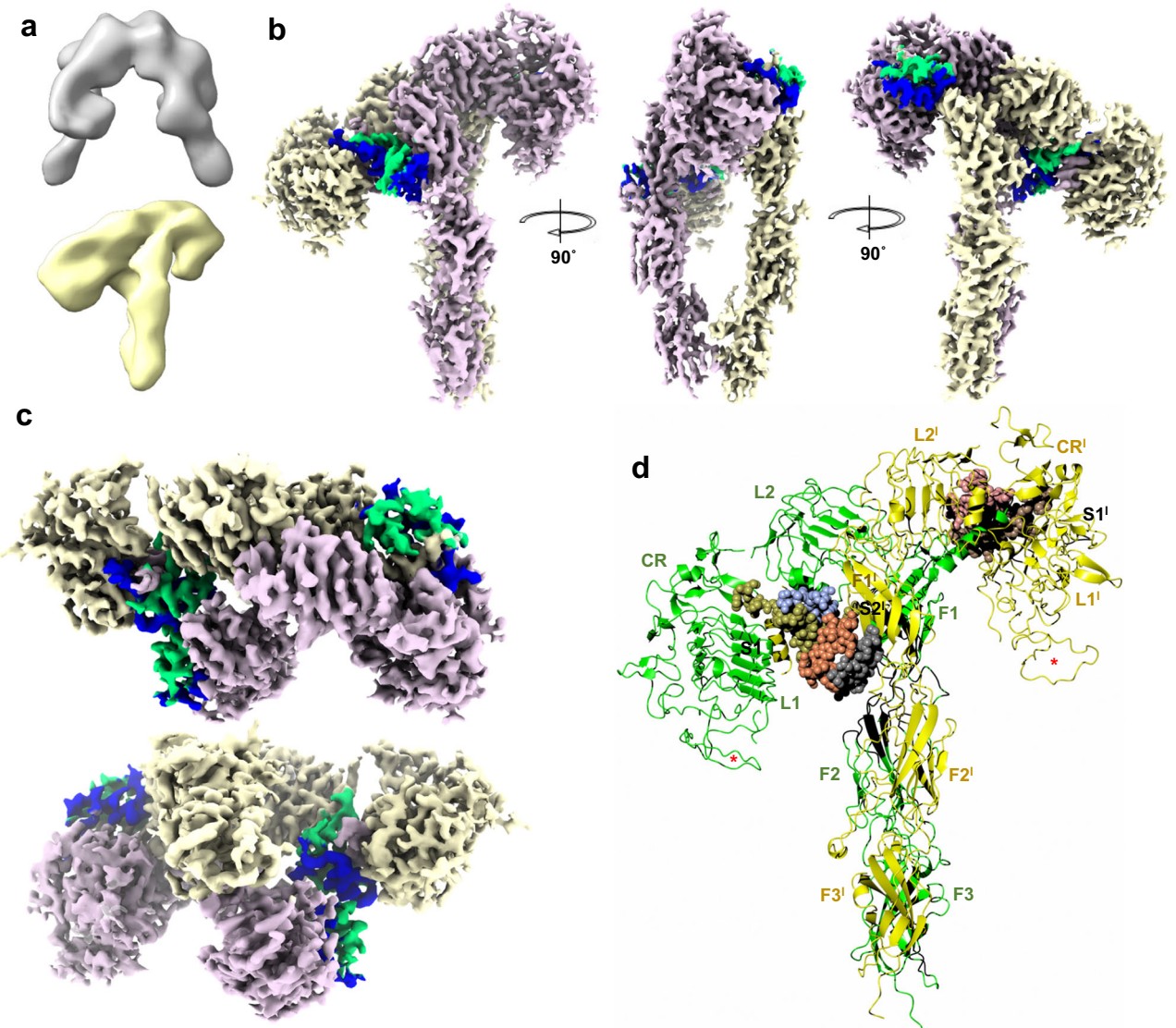

**Fig. 2 | CryoEM maps of the dmIR-ECD:DILP5 complex. a** The initial *ligand-free* (top/white) and dmIR-ECD:DILP5 complex (bottom/yellow) models from the classification. **b** The side views of the dmIR-ECD:DILP5 complex; the down 'dynamic' protomer is in yellow, and the top 'static' protomer in pink; DILP5 B-chains are in blue and A-chains are in green. Maps were sharpened by B-factor −50 Å² (see Methods). **c** The top and the bottom (from the end of the FnIII-3/3′ domains) views of the dmIR-ECD:DILP5 complex (above and below, respectively). **d** Ribbon

representation of the dmIR-ECD:DILP5 complex; domains are indicated by an abbreviated notation (e.g., FnIII-1 – F1), dynamic protomer - in green, static protomer - in yellow, DILP5s are depicted by spherical atoms, with the B-chains in gold, grey and brown, and the A-chains in blue, coral and pink. S1/1′, S2′ denote DILP5 binding sites 1/1′ and 2′, respectively. The red stars indicate the predicted model of the CR domain Gly491-Cys512 dmIR-specific insert which is not observed in the maps but included here to highlight its possible role in the dmIR.

---

The dmIR-ECD αβ protomers show the same L1-CR-L2-FnIII-1-FnIII-2-ID (α subunit) and ID-FnIII-2-FnIII-3 (β subunit) domains order and fold observed in h(m)IRs/hIGF-1R. These analogous blueprints are underpinned by a remarkable 3-D structural similarity of the individual domains of these receptors. The pairwise root mean square deviations (rmsd) between structures of the respective domains of dmIR-ECD and human receptors (calculated for the Cα atoms) are within the range of 0.80-2.15 Å (Supplementary Note 1). However, dmIR-ECD has several unique structural signatures - unique loops – which shed some light on how dmIR may handle signalling that involves seven DILPs through only one 'universal' dmIR (see further below).

The only significant deviation from the hIR fold is a large ~Gly491-Cys512 insert into the N-terminal region of the CR domain. However, the likely peripheral positioning of this loop - very disordered and not visible in the cryoEM map - suggests that it is not involved directly in hormone binding, or dmIR-ECD dimerisation, hence its role remains

unclear. It cannot be excluded though that it is involved in a firmer attachment of the L1 domain to the dmIR protomers in the *apo*-dmIR, or, just the opposite, that it prevents a fully down conformation of the L1-CR-L2 arm of the hormone-free protomer by clashing with the FnIII-3′ domain (Supplementary Fig. 4).

**Overall mode of DILP5:dmIR-ECD binding**

Despite a unique distribution of insulins in the three-DILP5:dmIR complex its asymmetric arm-down(Λ):arm-up(Γ) 3-D layout remains within the range of some known hIR asymmetric *T* conformers (e.g., PDB ID: 7PG0, 7PG2, 7PG4)[50]. The overall DILP5's binding modes in site 1 (down-arm (Λ) and up-arm site 1′ (Γ)) follows closely engagements of human insulins in respective h(m)IR sites (Fig. 2b–d). However, the third DILP5 molecule binds to site 2′ on the FnIII-1′ domain of the static protomer being adjacent to - and in contacts with - the down-arm site 1-bound DILP5, while site 2 is unoccupied (Fig. 2b–d, Fig. 3a, b). Such

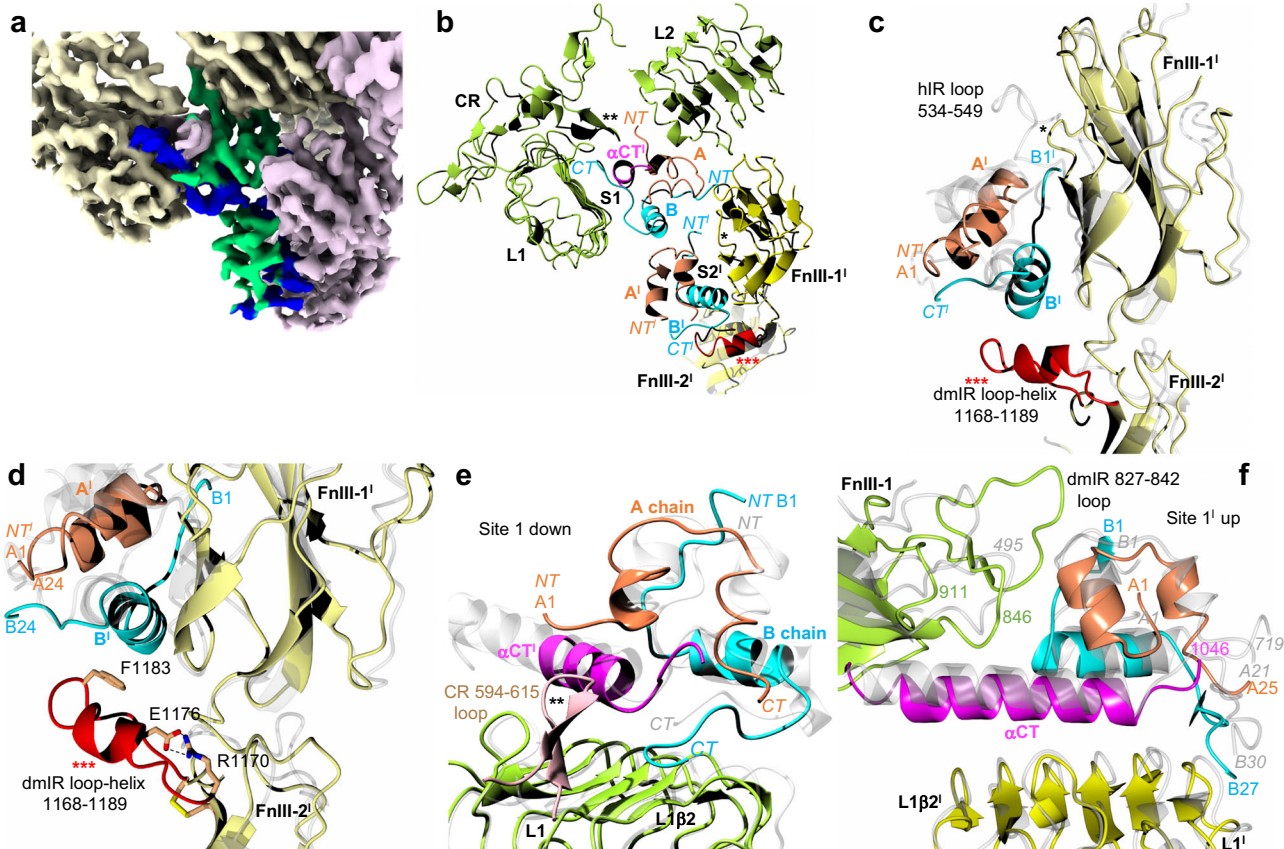

**Fig. 3 | dmIR-ECD:DILP5 binding sites. a** CryoEM reconstruction with model of DILP5 bound in site 1 and 2'; dynamic protomer - in yellow, static – in pink; DILP5 B-chains in blue, A-chains in green. **b** Ribbon representation of site 1-site 2' DILP5 binding regions; dynamic/arm-down protomer in green; static protomer in dark yellow, DILP5 B-chains in blue, A-chains in coral, α–CT' segment in magenta, (*) – FnIII-1' Lys884-Gly886 loop, (**) – CR Ala594-615Asn loop, (*** red) – FnIII-2' 1168Ala-1189Ser ledge helix-containing loop. **c** Site 2' in dmIR-ECD – colour coding as in (**b**), human insulin and hIR (PDB ID: 6SOF) in light grey (as in **d–f**). **d** A close-up on site 2' in dmIR-ECD complex showing the DILP5 supporting ledge from the FnIII-2' domain and putative HBs (dashed lines) that may stabilise this region; colour coding as in (**b, c**), N atoms in blue, O – in red. **e, f** Binding of the DILP5 in site 1 down and site 1'; colour coding as above; numbering in grey italic refers to hIR (PDB ID: 6HN5). Superpositions in (**c, d**) has been done on the respective FnIII-1 domains, and in (**e, f**) on the L1/L1' domains without the ligands (see details of the superposition targets in the Supplementary Note 1). Map in (**a**) was sharpened by B-factor −50 Å².

asymmetric insulin binding to the h(m)IR was postulated as structurally unfeasible due to envisaged steric clashes. Therefore this two-close-hormones-down and one-hormone up *T* asymmetric conformation of the DILP5:dmIR complex may represent a specific form of the dmIR, especially as the supersaturated insulin:receptor conditions yielded only *T* symmetrical h(m)IR conformers (e.g., PDB IDs: 6SOF, 6PXW/6PXV)[41,42].

### The conservation and uniqueness of sites 1/1' - 2' in dmIR-ECD complex

The overall features of DILP5 engagement with both sites 1/1' are very similar to the human insulin:site1 complexes, with a very low rms for both down-L1 (-0.9–1.4 Å) and up-L1' (-1.1–2.0 Å) domains, and respective helical α-CT segments (-0.8 Å) when compared with their hIR (and hIGF-1R) counterparts (Supplementary Note 1).

The overall folds of the dmIR and hIR FnIII-1 domains - the main components of site 2 and 2' − are also very similar (-1.2–1.9 Å rms range), and the overall mode of DILP5:site 2' interaction is - in general - similar in both receptors. However, DILP5 sits ~6 Å lower on the FnIII-1' side-surface than insulin in hIR's sites 2/2', with its B-helix also closer by ~2.7 Å to the FnIII-1', especially when compared at the Cα atoms of ValB16(LeuB17) (in brackets: corresponding sequences of hIR (or human insulin)) (Fig. 3c). This lowering of site 2' DILP5 is likely required for the simultaneous accommodation of both DILP5s in sites 1 and 2' in the lower-arm environment. The unique proximity of site 1/site

2'-bound hormones may be facilitated further by a shorter Lys884-Gly886 loop in dmIR (Pro536-Pro539 in hIR), and much longer in the dmIR Ala1168-Ser1189 loop (Ala785-Ser796) from the FnIII-2' domain that becomes now an integral part of site 2' (Fig. 3c, d). Here, the shorter Lys884-Gly886 FnIII-1 loop minimises the steric hindrance upon binding of both DILP5s into the lower arm site1/2' space, while the longer – dmIR specific - Ala1168-Ser1189 loop (with central short Ile1173-Thr1181 helix) provides a supporting ledge for the site 2'-bound hormone. This FnIII-2' ledge may be stabilised by putative Arg1170-Glu1176 salt-bridge, while Cys1169-Cys1188 (Cys786-795) disulfide at the base of this ledge contributes to the stability of this region, and ledge's Phe1183 can serve as a hydrophobic platform for DILP5 MetB11 (Fig. 3d). This contrasts with hIR where none of the FnIII-2 domains contribute to the respective sites 2/2'.

Both site 1 DILP5 binding modes reveals a similar pattern of the interactions in its 'classical' subsite 1a[42] which involves L1β2:α-CT':hormone interfaces. The helical α-CT' segment provides the same in-between anchoring base for DILP5, running relatively perpendicularly to the strands of the L1β2 sheet, as in the hIR (Fig. 3e, f). The structural conservation of DILP5 and human insulin binding in sites 1a/1a' maintains also the side chains' chemistry of these interfaces. For example, this can be seen: (i) on the L1β2:α-CT' surface − Leu368(Leu37), Phe416(Phe88), Tyr419(Tyr91), Val422(Val94), Arg446(Arg118), Glu448(Glu120), (ii) on the L1β2:DILP5 interface − Asp343(Asp12), Asn346(Asn15), Leu368(Leu37), Arg343(Arg65),

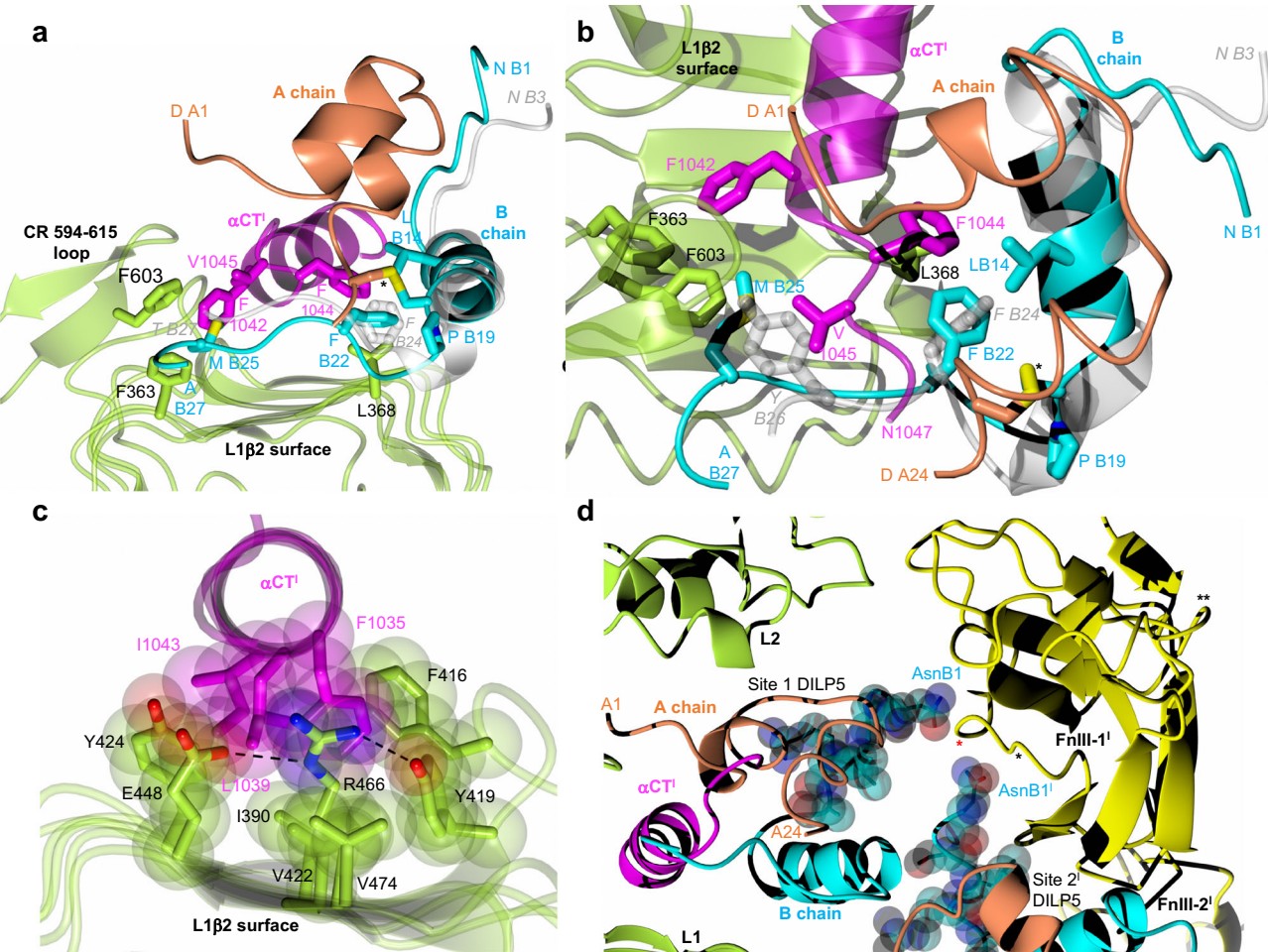

**Fig. 4 | Conservation of some *Drosophila* - hIR key interactions in site 1 and the arrangement of hormones' N-termini of the B-chains in site 1 and 2'. a, b** The 'side' and 'top' views of DILP5 in site 1 with some key hormone:dmIR-ECD interactions; (*) denotes A23-B18 DILP5 disulfide, colour coding as in Fig. 3, human insulin (PDB ID: 6HN5) in grey. **c** Examples of the preservation of the tethering of the α–CT' segment into L1-β2 surface of the L1 domain (site 1a); L1 domain and its side chains in green, α−CT' segment in magenta; oxygen atoms in red, nitrogen in blue, hydrogen bonds as dashed lines, side chains' van der Waals radii are also shown. **d** Convergence of DILP5s B1-B6 N-termini on the Pro880Pro881 region (indicated by a red star) of the FnIII-1' domain; B1-B5 atoms are shown as van der Waals spheres; the 884-886 and insert 827-842 loops are indicated by * and **, respectively; chains/protein colour code as in Fig. 3; a single amino acids code was used in (**a, b**) for image clarity; the superpositions in (**a, b**) has been done as in (**e, f**) in Fig. 3.

and (iii) on the α−CT':DILP5 interface − Phe1035(Phe705), Glu1036 (Glu706), Leu1039(Leu709), Asn711(Asn1041), Phe1044(Phe714), Val1045(Val715). Such conservation applies also to the DILP5 side chains involved in binding to site 1a/1a': GlyA4(GlyA1), ValA5(IleA2), ValA6(ValA3), AspA7(GluA4), TyrA22(TyrA19), AspA14(Asn21), PheB22(PheB24).

Nevertheless, the sequence differences between insect and human hormones somehow impact some important hormone: receptor interfaces in site 1a/1a'. For example, DILP5 (and DILP1) specific ProB19 substitution of human GlyB20, and deletion of the equivalent of human ArgB22, pushes the DILP5 B20-B26 chain closer to the L1β2 surface (Fig. 4a, b). Still, the conserved PheB22(PheB24) side chain is wedged into the hydrophobic cavity of PheI15(Phe714), Leu368(Leu37), LeuB14(LeuB15), CysB18(CysB19)-CysA23(CysA20), as PheB24 in the human hormone, and, remarkably, it seems that the B22(B24) phenyl rings occupy the same site in dmIR-ECD and hIR complexes (e.g., PDB ID: 6SOF, 6HN5) (Fig. 4a, b). Although the B23-B28 chain of DILP5 is different from the human sequence hence makes more dmIR-specific local interactions, it still follows the overall direction of this peptide across the L1β2 surface, as seen in human complexes. This, subsequently, allows the MetB25 − equivalent of the important human site 1a TyrB26 − to fill the Phe603, Val1045, Phe1042

hydrophobic cavity, alike structural role of TyrB26 in human complexes (Fig. 4b).

The preservation of the nature of such key contacts can be observed on other interfaces in dmIR-ECD. For example, α-CT' Phe1035(Phe705) is wedged into a hydrophobic cavity made by L1β2 and α-CT' side chains of Tyr419(Tyr91), Phe416(Phe88), Tyr424(-Phe96) and Leu1039(Leu79), (Fig. 4c). The environment of Phe1035 can also be stabilised by nearby Arg446(Arg118) involved in the dmIR-ECD/hIR conserved hydrogen bonds triad with Tyr419(Tyr91) and Glu448(Glu120), with a potential Arg446 guanidinium-π-Phe1035(Phe705) interaction as well.

The AspA1-PheA2-ArgA3 extension of the A-chain N-termini in the DILP5 C4 variant is readily accommodated in both sites 1. In site 1a (down), it is wedged between the CR-domain Ala594-Asn615 loop and L2 domain, and stabilised by ArgA3 CO-NH2 Asn606 and ArgA3 NH1 CO Glu660 hydrogen bonds. As the A1-A3 chain points outside of the dmIR, and is not obstructed by the receptor, it seems that the accommodation of DILPs with longer N-termini of their A-chains (DILP1, 6, 7) can be easily attained (Fig. 4a). The A1-A4 chain is very extended in site 1a, being more compact - more towards α-helical conformation - in site 1a'. Therefore, a shorter N-terminus of the A-chain of the DILP5 DB variant may be beneficial for dmIR-binding of

this more compact hormone ($K_D$ of 0.35 nM (0.76 nM for DILP5 C4 variant))[21].

The environment of the 1b sub-component of site 1, which involves hormone:FnIII-1 interface[42], is different in sites 1b and 1b' in dmIR-ECD due to a different display of parts of FnIII-1 to DILP5 on the down- and up-arms.

The upper (Γ) arm of the dmIR-ECD static protomer should have, in principle, an overall more classical site 1b' environment, but the human-like conservation of hormone-receptor contacts is rather uneven here due to a different length of corresponding FnIII-1 loops in hIR/hIGF-1R and dmIRs. Here, DILP5' engages its ArgB4(HisB10), Ala-B5(LeuB6), CysB6(CysB7)-CysA10(CysA10), ProB8 while FnIII-1 domain contributes Lys845, Asp846(Asp496), Pro847, and Arg848(Arg498). The dmIR-specific Thr827-Ile842 FnIII-1 domain insertion loop also impacts the structure of site 1b', as it may affect the conformation of DILP5 B1-B6 N-terminus in the static protomer (Fig. 3f), by narrowing the space for the B1-B6 chain that is still accommodated, and close to the DILP5 core in a tight T-like state. However, binding of DILPs with long N-terminal extensions of the B-chain, e.g., DILP1 and DILP7 (13 and 23 residues, respectively) can result in a steric hindrance with the 827-842 loop. Also, Pro8(SerB7) may have a particular role here by directing the B1–B7 chain as close as possible to the core of the hormone, to assure its tight fit between the hormone and the surface of the receptor. Such close packing of T-form-like B1-B7-chain is observed on all three DILP5:dmIR interfaces.

The site 1b interface on the Λ down-arm is less extensive, reflecting a different DILP5:FnIII-1' protein environment there (Fig. 3b). Instead of being exposed to FnIII-1' 'top' loops, the DILP5 faces here only a very small patch of this domain, with of AsnB1 hydrogen bonds with CO of Ser880 and NH of Ser881 as possible main contacts here. The 827-842 loop—important in site 1b'—does not have any structurally discriminatory role in site 1b, as this part of the FnIII-1' domain is away from the hormone binding site. Shrinking of the down-site 1b results also from the proximity of DILP5 hormones in sites 1a and 2'. The interface between these DILPs is between AspB12, MetB13, ArgB15 and ValB16 from hormone in site 1a, and SerB2, ArgB4, AsnA12-CysA14 region from site 2' DILP5. There is an interesting convergence of B1-B4 chains from these neighbouring DILP5 molecules on the Pro880-Pro881 patch of the FnIII-1' domain, which seems to serve as a hydrogen bonds centre for both B1/B1' Asn side chains (Fig. 4d).

The site 2' bound DILP5 uses mostly B-chain α-helix residues MetB11(ValB12), AspB12(GluB13), ValB16(LeuB17), AlaB17(ValB18)), and A-chain PheA16(LeuA13) and ArgA20(ArgA17) to interact with FnIII-1' β-strands (e.g., Ser815, Met882, Met888, Val889). Importantly, the site 2' hormone is expanded and supported by the dmIR-ECD specific Ala1168-Ser1189 ledge-like loop, where its central short Ile1173-Thr1181 helix runs below, and at ~45° to, the B8-B18 DILP5 α-helix (Fig. 3c, d). There is a lack of obvious, specific helix-helix interactions here, with the putative closest contact being between the Phe1183 benzene ring which may prop up hormone's MetB11 region.

Interestingly, site 2' DILP5 uses again a B-chain α-helix:β-strands protein-protein interaction motif, observed previously in its complex with the Imp-L2 binding protein[26]. However, despite the re-occurrence of similar DILP5 side chains on such interface (MetB13, ValB16, AlaB17, PheA16) its nature in the dmIR complex is different, as the B-helix runs almost parallel to FnIII-1' β-strands (Fig. 3c-d), while it is perpendicular to the direction of the β-sheet strands in Imp-L2.

Despite lack of human insulin-dimer stabilising triad (PheB24, PheB25, TyrB26 (Supplementary Fig. 1), dimerization of DILP5 was observed in more concentrated solution of this hormone[21]. However, its crystal-reported dimer (PDB ID: 2WFV) is stabilised by the two − antiparallel β-sheet-forming B-chains' N-termini AspB1-GlyB7 β-strands[21] − not the B20-B28 chains. Such arrangement is not found in the dmIR-ECD:DILP5 complex, where the interface between site 1 and

site 2' hormones is limited, and their B1-B7 termini are in an approximately parallel−practically contact free−arrangement.

## The role of Ala594-Asn615 (Cys259-280His) CR-domain loop

Although the CR domains of the hIR/hIGF-1R play a similar ligand-cavity enclosing role in both sites 1a and 1a', there is a distinct length and rearrangement of the Ala594-Asn615 (Cys259-His280) loop in the dmIR-ECD. It has a relatively broad conformation in the hIR/hIGF-1R, leaning more outside of the receptor and bound hormones, especially in the hIGF-1R where it points away from the C-domain of the hIGF-1. In the dmIR-ECD this loop is slightly longer, and—importantly—it is reconfigured into a narrow β-hairpin that points directly into the DILP5 binding cavity that would be occupied by the C-domain of single-chain insulin-like family of hormones (Fig. 3e, Supplementary Fig. 5). The space-occluding role of this loop can be amplified further by the N-glycosylation at Asn606, traces of which are indicated by the cryoEM reconstruction. The firm position of the Ala594-Asn615 loop is reinforced likely by hydrogen bonds of its central Asn606 to DILP5 ArgA3, and a contribution of loop's Phe603 to the hydrophobic cavity filled by DILP5 MetB25.

## Conformational ranges of the dynamic and static protomers

While comparisons of h(m)IRs/hIGF-1R structures are more direct due to a high similarity of primary-to-quaternary structures, the evaluation of dmIR-ECD in the context of human receptors is more complex, despite their overall similar domain organisation. Due to relatively lower sequence conservation combined with many of the reported h(m)IR structures being derived from a variety of constructs (i.e., with the presence or absence of transmembrane and intracellular regions or detergent micelles) that result in a broad spectrum of hormone:receptor affinities, it is not suitable to perform in-depth quantitative and functional cross-comparisons of hIR-ECD structures with the dmIR-ECD structure presented in this study. To simplify this process, we selected several reference dmIR-ECD Cα sites that are structurally - and frequently also sequence hIR-equivalents - hence they can serve as relative 'invariant' IRs structural pivots, which facilitate easier projections of the dmIR-ECD onto its h(m)IR and IGF-1R counterparts (see Supplementary Note 2).

In general, the mutual tolerance of site 1 and site 2'-bound insulins by the down-arm Λ protomer does not involve very special/radical angular movements of the L1-CR-L2 domains. Instead, it is achieved by conformational changes that are within the ranges observed for the static and dynamic h(m)IR protomers (Fig. 5). The trajectory movement of the L1-CR-L2 down Λ arm in the dmIR-ECD is not simply upward, but it combines up- and side-movements of these domains. Subsequently, the dmIR-ECD:DILP5 complex specific quaternary assembly is easily attained by structural changes that remains well within the broad spectrum of h(m)IR individual conformers[56] (see Supplementary Note 3, Supplementary Fig. 6).

## Impact of DILP5 binding on membrane-proximal domains

The binding of three DILP5s to dmIR-ECD sites brings together the FnIII-3/3' domains, despite the lack of any inter-protomers stabilising tags at their C-termini. The C-ends of FnIII-3/FnIII-3' domains are ~111 Å apart in the apo-hIR form (e.g. PDB ID: 4ZXB) to becoming ~16–20 Å close in some 1–3 insulin bound hIR conformers (e.g. PDB ID: 6HN4 + 6HN5, 7PG0-7PG4), and ~38 Å in hIGF-1R:hIGF-1 (PDB ID: 6PYH). In almost symmetrical T-shaped 4 insulins:IR complexes this separation is ~50 Å as the FnIII-3 C-termini (e.g., Lys917 in PDB ID 6SOF) lie 'outside' of these domains that, nevertheless, are in a close contact. In the dmIR-ECD this distance (measured at Lys1310) is ~15 Å. However, the close contacts (~4–5 Å) between the β-sheet hairpins of the FnIII-3/3' domains (Glu1242-Cys1258 (Glu846-Cys869 in hIR)) are not as symmetrical as in some insulin:hIR complexes, where they interact through the centres of their edges (around His858), or their tips (as in 4

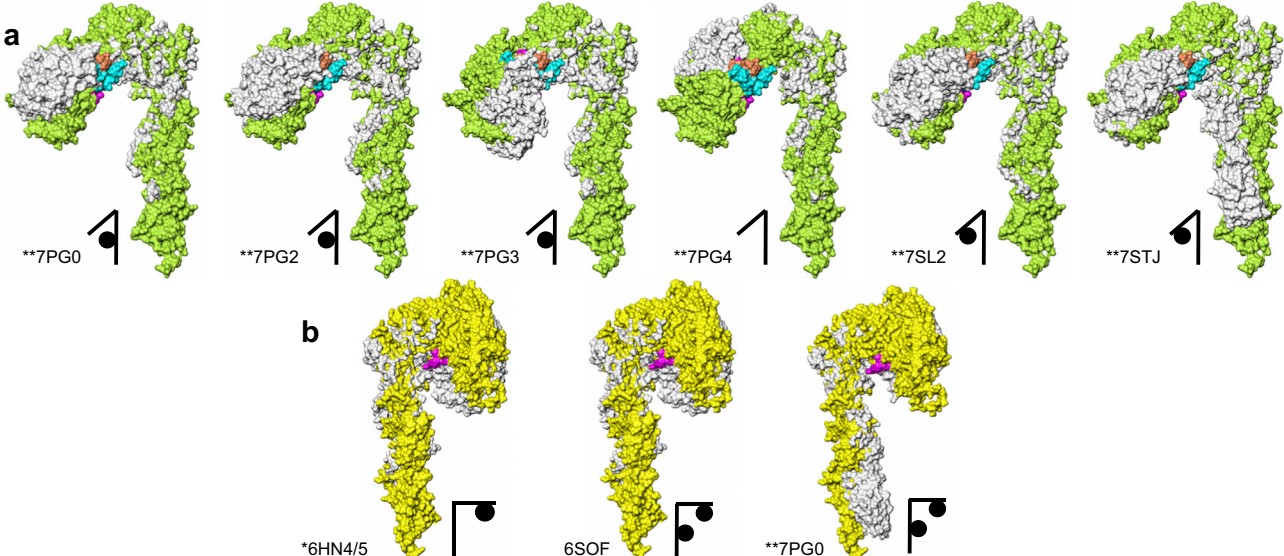

**Fig. 5 | Superposition of the dmIR-ECD:DILP5 dynamic and static protomers on the corresponding protomers of representative human insulin:IR complexes. a** dmIR-ECD dynamic protomers are in green and the static protomers in (**b**) are in yellow; human protomers are in white with their respective PDB IDs; DILP5 B-chain in blue, A-chain in coral; α-CT in magenta; black diagrams depict the number of insulins/protomer in the respective human complexes and the site of binding. All protomers were superposed in Coot (72) LSQ option on Cα atoms of the FnIII-1 domains (Ala807-Ans925 in dmIR on Glu471-Asp591 in hIR). The 827-847 and 881-887 loops in dmIR-ECD were removed prior to the superposition to minimise their potentially misleading bias in the superposition of a very similar cores of these domains (see Supplementary Note 1 for the exact targets). Two stars indicate ECD derived from the full-length IR models, one star−ECD from Leu-zipper containing ECD hIR, no star−ECD-only determined structure.

insulin:hIR, (PDB ID: 6SOF)). In dmIR-ECD:DILP5 complex this interaction falls between Glu1252' and Lys1257, reflecting a relative upright shift ( ~ 6-10 Å) of the static protomer. In general, the overall arrangements of the FnIII-3/3' domains in DILP5:dmIR-ECD is close to the one observed in one-to-three insulin/hIR complexes (PDB ID: 6HN4 + 6 HN5, 7PG0-7PG4).

## Modelling of DILP1-4 binding to the dmIR

In order to get an insight into possible binding of other DILPs to dmIR, models of DILP1-4 have been superimposed on DILP5 in site1/1' and 2' (see Methods). Although the cores of the DILPs should remain relatively structurally invariant, the N-/C-termini extensions of these chains are very different (SI Fig. 1). It seems that the binding of DILPs to the site 1a can be attained without any new steric hindrances; the extensions of DILPs A-chains (both N- and C-termini) and B-chain C-termini may be accommodated there without significant structural rearrangements of dmIR domains (Fig. 6a). These chains can also fit site 2' only after moderate structural alterations, as they project likely into an open, not dmIR-occluded space (Fig. 6b). However, this seems to be different for longer N-termini of the B-chains (e.g. in DILP1 and 7). They may have to thread in-between the hormones occupying sites 1-2' and the surface of the FnIII-1' domain, generating a particular DILP sequence-dependent, unfavourable interaction that may weaken the formation of site 2'. As the directions of B1-B6 chains of DILP5 in site 1 and site 2' converge on the Pro880-Pro881 region of the FnIII-1' domain, their extensions could clash with this part of dmIR. Similarly, the fitting of the long B-chain N-termini of DILP1/7 into the upper site 1b' of the static protomer can present a structural challenge, as these polypeptides would have to bypass the steric hindrance of the 827-842 FnIII-1 loop (Fig. 6c).

## Discussion

The conservation of insulin/IGFs signalling axis (IIS) in the animal kingdom - postulated by a wealth of biochemical records - is supported here by structural evidence at the insulin-like receptor level. Here we show that despite a large evolutionary gap some key molecular bases and principles of insulin:hIR (and to some extent hIGF1:hIGF-1R) interactions are remarkably preserved. This work expands the known conservation of the 3-D nature of human and DILPs onto: (i) analogous organisation and similar 3-D folds of the IR-ECDs (and − likely - their TKs[60]), (ii) similar nature of site 1/1' and site 2' hormone:IR coupling, and (iii) parallel blueprints of the overall quaternary conformations of the hormone:IR assemblies, where the dmIR-ECD:DILP5 complex falls well within the spectrum of structures observed for the h(m)IR, regardless whether it is compared with ECD-only or full-length human/mice receptors.

However, the dmIR-ECD possesses a few Dm-specific signatures. Here, the Ala1168-Ser1189 helix-containing loop - longer than its Ala785-Ser796 human homologue - provides a supporting ledge for the site 2'-bound DILP5 and makes this part of the FnIII-2' domain an integral part of hormone binding site. This FnIII-2' loop acts synergistically with the Lys884(Pro536)-Gly886(Pro549) FnIII-1' domain loop, which - being shorter than in h(m)IR/hIGF-1R - minimises the steric hindrance in the vicinity of the 2' site that is required for the simultaneous accommodation of spatially adjacent, and in close contacts, site 1/site 2'-bound hormones. These loops contribute likely towards an overall increase in stability of such arrangement of the hormones, unseen in human/mice insulin:IR and IGF-1R complexes. It is also interesting to note the differences in other FnIII-1 loops: (i) the Thr827-Ile847 insert into its human equivalent Pro495-Tyr492 loop region that may obstruct the escape of longer N-termini of DILP1/7's B-chains (especially in site 2' and 1'), and (ii) a shortening of dmIR Ala907(570Val)-Thr913(579Tyr) loop that liberates more space for the B-chain N-terminus. It can be speculated that the long Thr827-Ile847 FnIII-1 loop acts as a 'gatekeeper' which discriminates between DILPs with shorter or longer B-chain N-termini, thwarting an 'easy' binding of DILP1 or DILP7 to site 1'; this could restrict, for example, dmIR:DILP1 complex to two lower arm-like conformation, with a subsequent DILP1-specific intra-cellular signal. The variations of the dmIR's above-mentioned loops open possibilities for this single receptor to assimilate − with different affinities - six-seven different DILPs with their variable A and B-chain termini, and, subsequently, to acquire different quaternary structures with DILP-specific signalling outcomes.

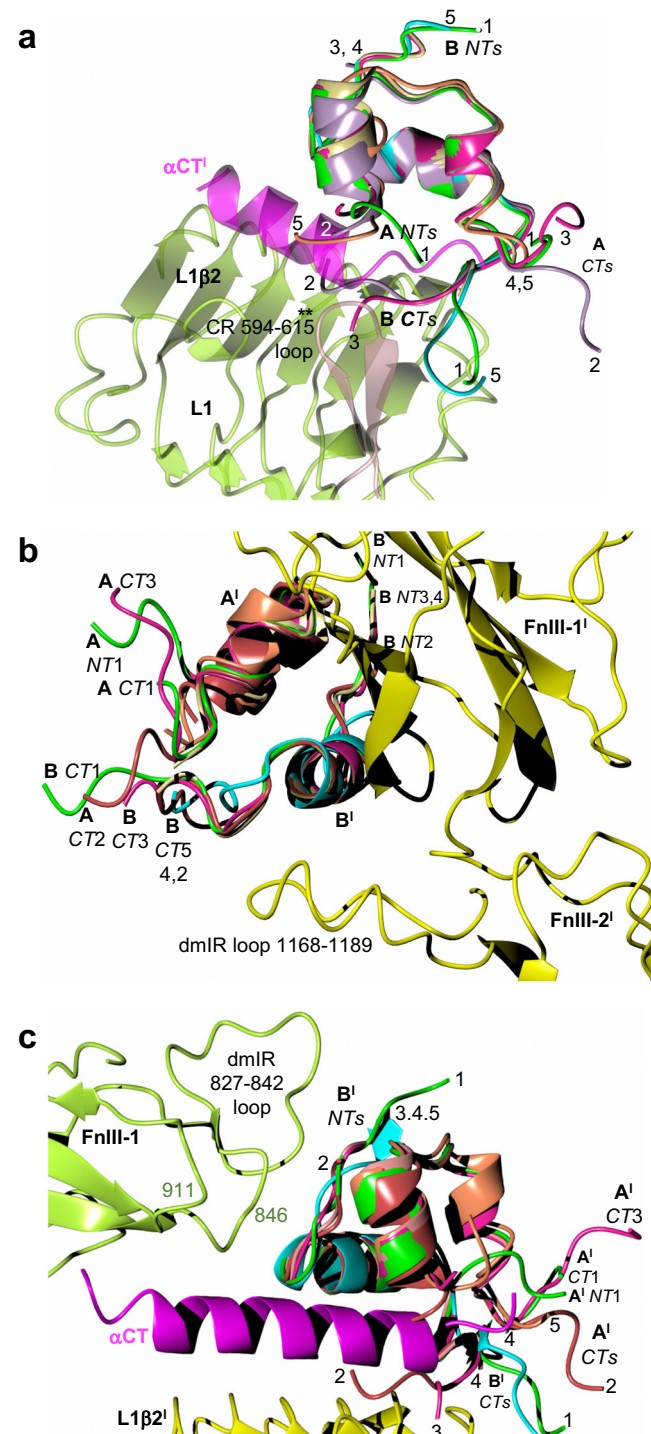

**Fig. 6 | Putative binding of DILP1-4 models in the dmIR-ECD. a** site 1a, (**b**) site 2′, and (**c**) the up site 1′. DILP5:dmIR-ECD colour coding as in Figs. 3–4; DILP1–lime green, DILP2–crimson, DILP–deep pink, DILP4–gold; *NTs-CTs* indicate the N- and C-termini of the respective chains, the numbers correspond to a particular DILP (e.g. 1–DILP1); DILPs' B-chains α-helices are superposed on DILP5 B8-B18 α-helix Cα atoms.

affinity of hIGF-1[21]. This $K_D$ is also higher - but much less - for hIGF-2: 30 ± 6 nM, indicating that hIGF-2 four amino acids shorter C-domain may be accommodated easier by the 594-615 CR loop. It is puzzling, whether such a remarkably low hIGF-1 binding affinity results entirely from its steric clash with the CR loop, or whether this loop prevents threading of the α−CT segment through the C-domain-encircled part of this hormone[36]. On the other hand, a relatively low ~60 nM $K_D$ for human insulin dmIR binding[21] may represent the impact of species-specific ILPs' sequence, if the core of the 3-D structural scaffold of these hormones is maintained in the process of speciation. Therefore, the Ala594-615Asn CR domain loop may be of evolutionary importance, as its less distinct structural role in the (m)hIR/hIGF-1R-like receptors results in insulin-like RTKs that are capable of binding a wider variety of two- or single-chain ILP-like hormones. This would allow the use of a relatively narrow population of insulin-like hormones (i.e., insulin, IGF-1/2) with a simultaneous assurance of a broad spectrum of signal transductions through the expansion of the hetero-dimerised isoforms of their IRs, e.g., the IR-A/IR-B and IR-A/IGF-1R hybrids in humans. It can be envisaged then that the presence of only one TK receptor in *Drosophila* required co-development of seven, C-domain free, DILPs to assure the necessary complexity of IIS axis in the insect life cycle.

Despite a low ITC-measured $K_D$ of 498 nM of the dmIR-ECD, the binding of three DILP5s to this construct brings together the FnIII-3/3′ domains which lack any inter-protomers stabilising tags at their C-termini or any IR-intracellular components. Similar conformation of these domains has also been observed in higher affinity (~30 nM) tag-free hIR-ECD 4-insulins T- and 3-insulins-*T*-state complexes[41] (PDB ID: 6SOF, 7QID). This supports further the correlation between hormone-coupling and convergence of IR-stem-protomers as an inherent feature of the ECDs in human and insect IR systems. The less symmetrical assembly of the transmembrane-proximal FnIII-3 domains in the dmIR:DILP5 complex may be considered as a part of more convoluted downstream translation of site 1/1′ and 2′ hormone-binding into not only a 'getting together' binary coupling of the respective cytoplasmic TKs, but their more complex translational/rotational 3-D movements. Such signal transduction would allow fine, sequential and more complex phosphorylation activation patterns of the TK kinases, hence different and modulated intra-cellular responses.

The occurrence of an asymmetrical *T*-like shape of the dmIR-ECD:DILP5 complex obtained at five molar excess of the hormone over the receptor may seem surprising in the context of mainly almost symmetrical T-shaped hIR:insulin complexes obtained at the saturated levels of the hormone. However, the structural asymmetry of dmIR-ECD:DILP5 complex can be considered in the context of the negative cooperativity in this family of receptors[1,7,28,29,56,61,62], which has been postulated to be linked with the ligand-induced IR asymmetry[30]. In the classical kinetic assays of ligand-accelerated dissociation of a bound radioligand, the dose–response curve for insulin dissociation is bell-shaped with disappearance of its acceleration at concentrations over 100 nM[61,62]. In contrast, both the hIGF-1R at the hIGF-1 supraphysiological level[62], and the dmIR at the excess DILP5[21], maintain the negative cooperativity, showing indeed its correlation with the asymmetrical configuration of their complexes. This is also supported by the data with IRs and IGF-1Rs with modified α−CT segments[45]. It should be noted though, that the DILP5:dmIR-ECD complex reported here is based on a low-affinity ectodomain, lacking also any FnIII-3 domains association-inducing, or stabilising, tags. These factors alone may play a role in trapping this ECD in only three hormone-receptor conformation. Therefore, whether such hormone:receptor asymmetry is species-specific conformational barrier of the dmIR-ECD, or whether it may be pushed further into the symmetrical T-state by other types of DILPs in the negative cooperativity-free process, requires further studies. Moreover, the DILPs-induced allostery of the dmIR can diverge even further from the human models, as this receptor has only two of

Another characteristic structural feature of the dmIR-ECD is the reshaped - and slightly longer than in the hIR - Ala594-615Asn (Cys259-280His) CR-domain loop, which may prevent binding of any insulin-like hormone with a significant C-domain (e.g., hIGF-1-like). Indeed, the $K_D$ of hIGF-1 for dmIR high-affinity site 1 is 5913 ± 32 nM while that of DILP5 C4 is 0.76 ± 0.16 nM, indicating a 7780 times lower

human three important protomers-linking cysteines in the ID segments (Cys1010, Cys1012), with human third Cys682 being replaced here by Tyr1009 (this ID regions are not observed in the maps). Also, the structural/functional impact of the dmIR unique ~60 kDa addition of the CD domain to the tyrosine kinase part of this receptor cannot be excluded.

The DILP5:dmIR-ECD structure also opens a new venue for specifically dmIR engineered—and phenotypically validated—transgenic *Drosophila* flies as models for particular human diseases. Here, for example, our results provide a rational explanation of the impact of the recently reported Tyr902Cys missense mutation in the dmIR that yields temperature-sensitive flies with broad metabolic effects, making this mutant a good Type 2 Diabetes model[63]. Tyr902 is at the centre of a hydrophobic core at the base of the 867-877 loop that spans FnIII-1/FnIII-1′ domains of the inter-protomers α-subunits via the Cys873-Cys873′ disulfide bridge. The Tyr902Cys mutation can destabilise this region (crucial for the dimeric state of the IR) by, for example, an abnormal intra-protomer Cys902-Cys873 disulfide coupling, as suggested by Banzai and Nishimura[63] (Supplementary Fig. 7), leading subsequently to a low folding efficiency of dmIR hence its reduced bioavailability. Moreover, another temperature sensitive and metabolic phenotype Val811Asp mutation[63] can also be explained by the destabilisation of the Val811 hydrophobic cavity, which is in an immediate proximity of Tyr902-containing β-strand.

In summary, this report provides the evidence of the structural conservation of the IIS axis in Metazoa at the IR level, underpinning the wealth of biochemical and cellular data postulating an overall similar functional mimicry of insulin signalling in humans/vertebrates and non-vertebrates. The dmIR-specific structural signatures contribute also to a better understanding of *Drosophila* as a model for human pathologies and aging, highlighting possible dmIR-targets for rational genetic and cellular manipulation of this system.

## Methods

### Production of dmIR-ECD
The construct used in this study was subcloned from a vector encoding a codon optimised version of Drosophila insulin receptor (Uniprot ID: P09208) ectodomain (a gift from Nikolaj Kulahin Roed (Novo Nordisk). The coding sequence beginning with the native signal sequence of dmIR (264H-NYSYSPGISLLLFILLANTLAIQAV-290V) was predicted using the SignalP 6.0 web service[64]. DNA encoding the amino acids 264-1309 of dmIR was subcloned into the vector pBAC™4x-1 DNA (Novagen) downstream of the p10 promoter with the addition of a starting Methionine and C-terminal StrepII-tag. Bacmid (gift from Ian M. Jones, University of Reading) was purified from *E. coli* and digested with BsuI[65]. V1 baculovirus was produced by mixing linearised bacmid with purified baculotransfer vector and FugeneHD transfection reagent (Promega) and incubated with adherent Sf9 cells (ThermoFisher, cat. no. 11496015) at ratio of 3:4:12 bacmid:vector:FugeneHD in a 6-well plate (0.75 microgram: 2 microgram: 6 μL per 2 mL well). V1 to V2 baculovirus amplification was performed in Sf9 suspension culture, and the GFP present in the baculotransfer vector, upstream of a polyhedrin promoter, was used to determine optimum amplification before harvest (typically >90% cells fluorescent).

### Expression and purification of dmIR-ECD
dmIR-ECD was expressed in Sf9 cells infected with V2 baculovirus at a MOI > 1 and infection followed using the GFP marker to determine optimum time for harvest (typically 72 h, with >90% cells fluorescent). Conditioned media (2.4 L) was cleared by centrifugation at 500 x *g* for 20 mins at 4 °C followed by a second clearing of debris by centrifugation at 5000*g* for 30 min at 4 °C. The cleared conditioned tangential flow media was placed onto a 30 kDa cut-off filtration (TFF) column (Repligen, S02-E030-05-N, surface area 790 cm²). The media

was concentrated 10-fold by placing the feed tube in the media and applying 12 PSI transmembrane pressure (TMP) and 20 mL/minute flux rate. The retentate was diafiltrated by placing the feed tube in the purification buffer until at least a 10-fold volume of the purification buffer was consumed. The diafiltrate was incubated with a $CaCl_2$ and $NiCl_2$ solution stirring at RT for 30 minutes and cleared of debris by centrifugation at 18,000 x *g* for 60 minutes at 4 °C. Samples were passed through Strep Trap HP 1 mL column (Cytiva), washed with the purification buffer and eluted with 2.5 mM Desthiobiotin. The eluent was pooled and concentrated to a volume of ~0.5 mL with a 30-kDa Vivaspin concentrator (GE Healthcare) and further purified by size-exclusion chromatography (SEC) with a Superdex S200 10/300 column (Cytiva) in SEC buffer (20 mM HEPES, 200 mM NaCl, pH 7.4). dmIR-ECD containing fractions were concentrated to 2 mg/mL with a 30-kDa Vivaspin concentrator. Purity of samples was assessed by SDS- and Native-PAGE (Supplementary Fig. 8b-d).

Titration of DILP-5 hormone into dmIR-ECD was carried out using MicroCal iTC200 instrument (Malvern Instruments). Titration data obtained from nineteen 2 μL injections of DILP5 5803 g/mol (200 μM) into dmIR-ECD 233538 g/mol (5.3 μM). All injections were carried out in 50 mM Tris, 150 mM NaCl, pH 7.4 buffer. Data was fitted using "One set of sites" binding model. Data fitting and analysis was carried out with the manufacturer's software, MicroCal PEAQ-ITC Analysis (Supplementary Fig. 8e).

### Cryo-electron microscopy data collection of dmIR-ECD complex
The dmIR-ECD:DILP5 complex was prepared by incubating equal volumes of 2.3 mg/mL (10 μM) dmIR-ECD and 0.3 mg/mL (50 μM) of DILP5 (gift by Novo Nordisk) in 50 mM Tris, 150 mM NaCl, pH 7.4 buffer, resulting in a five-molar hormone excess during making of this complex. It was subsequently prepared on UltraAuFoil R1.2/1.3 gold support grids (Quantifoil). 3 μL of sample was applied to glow-discharged grids, blotted for 2 s with −10 force and vitrified by plunging into liquid ethane using the FEI Vitrobot Mark IV at 4 °C and 100% relative humidity. Micrographs were collected at the Diamond eBIC facility on Titan Krios microscope (FEI) operating at 300 kV, and equipped with K3 camera and an energy filter (Gatan) using slit width of 20 eV. Automated data collection was performed using FEI EPU software. 4858 movies were collected in super resolution mode with super resolution pixel size of 0.4145. The defocus range chosen for automatic collection was 0.5 to 4 μm.

### Image processing and reconstruction
All datasets were processed in RELION 3.1.2[66] and Topaz 0.2.3[67]. Movie frames of the dmIR-ECD:DILP5 complex were motion-corrected binning by 2 to the physical pixel size of 0.829 Å, and dose-weighted using a dose per frame of 1.03 e/Å² using the Motioncorr2 program[68] (see Supplementary Fig. 3, Supplementary Fig. 9, and Supplementary Table 1 for details). CTF parameters were estimated using CTFFIND4[69]. An initial round of autopicking was performed in RELION using LoG. 1.2 million particles were extracted and subjected to several rounds of reference-free 2D classification to remove particles associated with noisy or contaminated classes resulting in 318,846 particles. 2D classes showing sharp structural features were chosen to build an initial 3D model. This initial model was then used for 3D classification. The class showing well-defined structural features was then selected for 3D refinement which gave a reconstruction with a resolution of 9 Å. 1250 particles from this class were then used to train Topaz on a dataset comprising all micrographs. Following picking with Topaz using this model ~730 K particles were extracted from which rounds of 2D classification and a selection for the initial model resulted in two reconstructions consistent with 'open' and 'closed' conformations of the receptor. Further rounds of 3D classification and 3D refinement resulted in a reconstruction of the dmIR-ECD:DILP5 complex at 4.0 Å resolution.

## Model building refinement and validation

Individual domains of dmIR-ECD were predicted using AlphaFold/ 2.0.0-foss-2020b[70] and docked as rigid bodies into the 4.0 Å resolution cryoEM maps using UCSF Chimera's "Fit in map" function[71]. The AlphaFold predictions were very accurate, with small rms deviations between the model and final structure domains (0.43, 0.68, 0.48, 0.53, 0.54 and 0.52 Å for L1, CR, L2, FnIII-1/2/3 respectively (Cα atoms LSQ superposition in Coot[72]). The CR domain has been predicted with the lowest confidence. Atomic model building of the DILP5 hormone was performed using the previously reported crystal structure core residues of DILP5 as an initial model[21], which was docked as a rigid body as described above (Supplementary Fig. 10). This initial model was modified using Coot[72]. CryoEM maps were sharpened for model building and figure preparation by the B-factor of −50 Å$^2$ using Coot cryoEM tools[72]. Interactive remodelling/refinement of the model into cryoEM maps was performed using ISOLDE[73]. Real-space refinement was carried out with secondary structure restraints using Phenix[74]. Model geometries were assessed with MolProbity[75] (Supplementary Table 1). Structures and maps in the figures were rendered with PyMOL (http://www.pymol.org/) or ChimeraX[71], and with 0.0288 contour level. Figures were also made with the CCP4mg programme[76].

## Modelling of DILP1-4

Structures of DILP1-4 were homology modelled in Modeller version 9.21, using cryoEM structure of DILP5 from its complex with DmIR. A total of 100 models were generated using the cysteine disulfide bridging constraints implemented in the software. Models with the least DOPE scores were used for further analysis. protein-protein interaction studies were performed using the ClusPro server[77]. DILP5 Cα atoms 8-18 spanning B-helix in site1/1' and 2' were targets for the LSQ superposition in Coot, with the corresponding regions in the other DILPs: DILP1 B22-B32, DILP2 B5-B15, DILP3 B7-B17 and DILP4 B7-B17.

## Reporting summary

Further information on research design is available in the Nature Portfolio Reporting Summary linked to this article.

## Data availability

The atomic coordinates of the DILP5:dmIR-ECD generated in this study and associated cryoEM reconstruction have been deposited in the Protein Data Bank and EM data bank with the accession codes 8CLS and EMD-16718, respectively.

The following already published structures were used in this report: 6SOF, 6HN4, 6HN5, 7PG0, 7PG2, 7PG3, 7PG4, 6PXW, 6PXV, 4ZXB, 6PYH, 7SL2, 7STJ, 7SL7, 7STK, 7S0Q, 7MQS. Source data are provided with this paper.

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

## Acknowledgements

The authors thank Maria Chechik, Oliver Bayfield and Herman Fung for the help in the initial CryoEM stages of this project, Jamie Blaza for help during the project, Tony Wild, and the Wolfson Foundation for funding the Eleanor and Guy Dodson Building and associated cryoEM facilities. We also wish to acknowledge the work of Johan Turkenburg and Sam Hart for coordinating the YSBL CryoEM Facility. We are grateful to Gerd Schluckebier and Nikolaj Kulahin Roed (Novo Nordisk A/S, Maaloev, Denmark) for the dmIR-ECD clone and DILP5 protein. Novo Nordisk is greatly acknowledged for funding of the G. G. Dodson Fund in York. C.V. and A.M.B. were supported by MRC Grant MR/R009066/1, O.F. was funded by the G. G. Dodson Fund in York sponsored by Novo Nordisk A/S (Maaloev, Denmark), T.S. and A.M.B. were also supported by the BBSRC Grant BB/W003783/1. We acknowledge Diamond for access and support of the Cryo-EM facilities at the UK National Electron Bio-Imaging Centre (eBIC), proposal BI28576, funded by the Wellcome Trust, MRC and BBSRC.

## Author contributions

C.M.V. designed and conducted experiments, prepared dmIR-ECD samples, analysed data solved CryoEM structure, helped with writing and editing of the manuscript, O.F. helped with dmIR-ECD sample preparation, H.T.J. helped with CryoEM structure elucidation, T.S. was responsible for the modelling work and helped with discussions, P.D.M. analysed the results and worked on the manuscript, A.M.B. designed the study, analysed the results and wrote the manuscript.

## Competing interests

The authors declare no competing interest.
