## [Peer Review File · Nature Communications]

REVIEWER COMMENTS

Reviewer #1 (Remarks to the Author):

Overall, the presented manuscript is of major importance and a major milestone towards comprehensive understanding of IR function. It provides clear evidence that the recently published breakthrough publication/s on IR activation and the number of insulin binding sites is evolutionary conserved. Moreover, this manuscript paves the way towards a breakthrough in both, developmental biology and cellular metabolism, as it was and still remains entirely cryptic as to why *Drosophila* is producing the many kinds of different DILPs.

I would like to applaud to the authors. Nature Comm is the perfect journal for publishing this article.

Major points:

It is not clear how the authors determine the resolution of their final reconstruction to be 3.9Å (see results and material and methods). The usual FSC report for the independent half-maps of the final reconstruction used to evaluate the approx. global resolution is missing from the manuscript. In the included pdb report showing quality indicators for both the final reconstruction as well as the model the authors indicate a resolution of 4.00 Å (see e.g. first page). This is in contradiction to the analysis of the FSC under point 8 of the pdb report. Here both the pdb calculated as well as the author provided FSC curves indicate an approx global resolution estimate according to the "gold-standard" resolution cut-off criterion of 0.143 (Rosenthal & Henderson, 2003) of 4.5 Å and 4.6 Å respectively. This has significant consequences for the interpretation of the model and the manuscript.

At an apparent resolution of 4.5 Å side chain level interpretation of structural data becomes highly speculative.

This means that many of the structural claims (e.g. all of Figure 4) are if at all only weakly supported by the data and highly speculative at best.

The overestimated resolution could also explain why the 3D maps in Figure 2 and 3 appear so noisy and show signs of clear over-refinement (streaky artifacts)

Minor points:

The deposited model has 4.2% Ramachandran outliers indicating a geometrically unsound model (see page 2 of the pdb report, a "good" model usually has >0.1% Ramachandran outliers).

This puts a further question mark on the structural data and claims made by the authors.

It is not clear from the data shown in Figure 3A how the authors were able to determine orientation/fits of the DILP5 promoters into the respective densities

SI Figure 3 - initial model after 698k 2D class selection: second model appears to be apo-IR - have the authors pursued this further?

please include a customary tableS1 outlining the dataset characteristics, characteristics of the final 3D reconstruction and the model quality indicators. For a template please refer to other recent SPA cryoEM publications.

Please include a half map FSC and a model vs map FSC for the final reconstruction/model as well as an angular sampling plot in the figures. This is customary and helps readers to evaluate the quality of the reported reconstruction and the fit between model and map.

The contour level provided by the authors in the pdb report appears off compared to the level used in the paper

The authors must expand the the method section and provide more details, such as the exact ratios of receptor ligand, to name one incomplete information.

Reviewer #2 (Remarks to the Author):

Manuscript NCOMMS-23-07694:

Structural Conservation of Insulin/IGF Signalling Axis at the Insulin Receptors Level in *Drosophila* and Humans.

Viola et al.

The manuscript reports the cryo-EM structure of ectodomain of *Drosophila melanogaster* insulin receptor (dmIR) in complex with three instances of *Drosophila* insulin-like protein 5 (DILP5). On the whole, I find the results compelling and the discovery of a novel interaction between the site 1 and site 2' DILP5 and between the site 2 DILP5 and domain FnIII-2 is intriguing. The authors findings add to the repertoire of conformations and stoichiometries observed for liganded human IR, human IGF-1R and their hybrids, underscoring the need for clear interpretation of how these structures fit into physiological signalling scenarios.

As such, I find the manuscript suitable for publication in Nature Communications, but there are a number of shortcomings that need to be addressed prior to acceptance for publication. In no particular order, these include

1. The manuscript has many typographic, spelling and grammatical errors and inconsistencies (suggesting to me that the manuscript may have been written in haste, perhaps in the face of competition). I recommend that the authors undertake a thorough proof-reading, as well as a check of the manuscript for self-consistency of style.

2. In the Abstract:

(i) "dmIR binds three DILP5 molecules in an unseen arrangement": whereas what this means may become clear as the manuscript unfolds, it is not clear in the context of the abstract alone. It could mean that the authors don't observed the three DILP5 molecules at all! I suspect that the authors mean "hitherto-unseen" rather the "unseen"?

(ii) "revealing unique structural signatures" is likely an over-statement. "Unique" with respect to what? The reported structural signatures may well occur across many insect species.

(iii) "provides structural proof of the evolutionary conservation..". Again, I would say this is an over-statement. I would be happier if it was worded "adds structural support to the evolutionary conservation...".

3. I may have missed it, but the authors make no reference to the N-terminal residues of dmIR that were excluded from their construct (which begins at residue 292). Where precisely does the signal peptide lie? Even so, there remains a significant N-terminal "tail" prior to domain L1 - is there any indication of ordering of these residues within the map? Could they play a role in modulating either ligand binding or the structural conformations accessible to the receptor?

4. The reported cryoEM methodology and statistics need to be enhanced. The authors should

(i) include an FSC plot to convince the reader that the 3.9 Angs resolution has been correctly reported. In particular, I note some discrepancy between the resolution estimates contained in the PDB report and that stated in the manuscript.

(ii) provide representative electron density at the residue level at selected locations within the structure in order to convince the reader that the resolution is as reported. Figures 2a and 3a do not suffice in this regard.

(iii) provide within the main figures a panel that indicates the putative (low-resolution) 3D conformation of the apo receptor extracted from the 3D classification (and indicate the associated class clearly within Figure S3).

(iv) whereas the PDB report contains information about model quality, the manuscript should also include at least some of this detail (e.g., Ramachandran statistics).

5. The "extra" loop within the CR domain of dmIR is intriguing and, although not within the same location, is reminiscent of the additional loop within the CR domain of human EGFR that mediates dimerization of liganded EGFR. Is it possible for the authors to speculate on whether extra CR loop in dmIR could play a role in receptor-receptor interactions of either the liganded or ligand-free dmIR?

6. At the beginning of the Results section, it is important for the authors to provide a brief summary of precisely what protein construct is being analysed, how it was expressed and purified and a statement that it was assessed as being "fit-for-purpose".
7. There needs to be consistency in cited four-letter PDB IDs - the ms currently uses a mixture of upper and lower case.
8. SI Table 1. The inclusion of four decimal places in the r.m.s.d. values way exceeds their likely accuracy.
9. Figure 3c-f. The authors need to state what is being overlaid with what in generating the respective dmIR:hIR superpositions. Are the overlays based on the receptor domains alone (and if so which) or do they include the ligand(s)?
10. In the Figures and throughout, there are multiple versions of the prime symbol (') or ('). These should be standardized, my preference is for the former.
11. In Figures, the central beta sheet of domain L1 referred to as β 2L1, whereas in the text it is L1 β 2.
12. The change in relative conformation of the domains in the receptor family upon ligand binding is likely correlates with the degree of association achieved between domains FnIII-3 and FnIII-3'. The authors report on p14 that the association between these domains is asymmetric, but no detail is given. Can the authors please clarify if these domains are in contact and, if so, via which elements, and does their association mimic any associations seen in liganded hIR and/or hIGF-1R? (note two typos in the associated para on p14 regarding PDB IDs: should be 4ZXB and 6HN4+6HN5).
13. Fig 1d: are the sequence similarity numbers in the Figure percentages or number of conserved residues? This is not stated in the caption and is not clear.
14. The authors need to include the missing reference in the caption to SI Figure 7.
15. On p17: " lacks any FnIII-1 domains joining – e.g. Leu-zipper-like tags ", the authors presumably mean "lacks any FnIII-3 domain joining tags"??

16. The using of AlphaFold. No literature reference is given in the ms or in the Reporting Summary to this software or to its version number. Given the relative newness of AlphaFold(1 or2?), it would be opportune for the authors to comment on the perceived accuracy or otherwise of the AlphaFold prediction when assessed against the cryoEM maps.

17. The authors may also discuss the propensity or otherwise for DILP oligomerization (i.e., with reference to that of human insulin), as it may well be that differences in association here with respect to the insulin:hIR system may be linked to this. In particular, DILPs lack the aromatic triplet that mediates human insulin dimerization.

18. p8, reference to SI Table 2 in 2nd para. I believe that this reference is correctly to SI Table 1.

19. SI Fig. 8E. The ITC trace is marked as "typical" (which means more than one experiment was conducted). Why were these not all included to yield a mean $K_d \pm SD$? What type of analysis was conducted to reach the stated value of 274 nM (one-site or two-site, etc.)?

20. SI Fig 7. There has been uncertainty regarding the pairing of the inter-alpha-chain disulfides at the Cys triplet in the insert domain of hIR. I note from SI Figure 2 that dmIR contains only two Cys residues at this point, suggesting a different structural motif at this site. Does the authors' map allow resolution of these bridge(s)?

I recommend publication of the manuscript subject to the above revisions being made to the journal's satisfaction.

REVIEWER #1

General Response: We are very grateful to the Reviewer for a very constructive critique. All his points help us to weed out technical deficiencies, getting the paper into fully professional shape.

Overall, the presented manuscript is of major importance and a major milestone towards comprehensive understanding of IR function. It provides clear evidence that the recently published breakthrough publication/s on IR activation and the number of insulin binding sites is evolutionary conserved. Moreover, this manuscript paves the way towards a breakthrough in both, developmental biology and cellular metabolism, as it was and still remains entirely cryptic as to why Drosophila is producing the many kinds of different DILPs. I would like to applaud to the authors. Nature Comm is the perfect journal for publishing this article.

Major points:

(1) It is not clear how the authors determine the resolution of their final reconstruction to be 3.9Å (see results and material and methods).

The usual FSC report for the independent half-maps of the final reconstruction used to evaluate the approx. global resolution is missing from the manuscript. In the included pdb report showing quality indicators for both the final reconstruction as well as the model the authors indicate a resolution of 4.00 Å (see e.g. first page). This is in contradiction to the analysis of the FSC under point 8 of the pdb report. Here both the pdb calculated as well as the author provided FSC curves indicate an approx global resolution estimate according to the "gold-standard" resolution cut-off criterion of 0.143 (Rosenthal & Henderson, 2003) of 4.5 Å and 4.6 Å respectively. This has significant consequences for the interpretation of the model and the manuscript.

At an apparent resolution of 4.5 Å side chain level interpretation of structural data becomes highly speculative.

This means that many of the structural claims (e.g. all of Figure 4) are if at all only weakly supported by the data and highly speculative at best. The overestimated resolution could also explain why the 3D maps in Figure 2 and 3 appear so noisy and show signs of clear over-refinement (streaky artifacts).

Response: The resolution of 4.0 Angstrom reported in the manuscript was determined by Relion during postprocessing with a mask. We also wondered why the PDB Deposition validation reported 4.5 and 4.6 Angstrom resolution, and now realise that it was due to uploading an xml file derived from unmasked half maps. This has been corrected so that the xml file of the masked map is now included in the deposition. Additionally, there is now an FSC plot overlay of the unmasked half maps and masked maps in the **SI** (Fig. 9, p.13) that we hope will address this concern.

(2) Minor points:

(i) The deposited model has 4.2% Ramachandran outliers indicating a geometrically unsound model (see page 2 of the pdb report, a "good" model usually has >0.1% Ramachandran outliers).

This puts a further question mark on the structural data and claims made by the authors.

Response: Since the reviewer's helpful inspection of our structure, a significant number of

model Ramachandran outliers have been fixed using Coot and ISOLDE and refined using PHENIX real space refinement tool. The Ramachandran outliers are now at 0.28% (see new **SI** Table 3, p.15). The new model has been uploaded to the PDB Deposition site. All these corrections concerned residues in areas where the map was difficult to interpret and did not involve hormone binding, important interfaces, etc. They did not change or affect any results, interpretation or conclusion.

(ii) It is not clear from the data shown in Figure 3A how the authors were able to determine orientation/fits of the DILP5 promoters into the respective densities

Response: The figures (2, 3a) have been reanalysed and new figures produced aided by sharpening of the cryoEM map, and more suitable contouring in Chimera. They indicate unambiguous location of the hormones and important parts of the dmlR-ECD.

(iii) SI Figure 3 - initial model after 698k 2D class selection: second model appears to be apo-IR - have the authors pursued this further?

Response: Two initial models could be determined from particles picked using the neural network picker TOPAZ. The 'closed' holo-model was pursued in this study since the sample was mixed with the excess of the hormone. The second model that appears to be apoIR has significantly fewer particles and resolution is much lower >8.5 angstrom and therefore not included in this study. Nevertheless, there is a new figure 2a that shows the initial classes for apo and holo dmlR-ECD. We pursue the apo-dmlR issue; this – current – work is the first year of our BBSRC grant; hopefully we will be able to report it later on this year.

(iv) please include a customary table S1 outlining the dataset characteristics, characteristics of the final 3D reconstruction and the model quality indicators. For a template please refer to other recent SPA cryoEM publications.

Response: A customary table of dataset characteristics and structure and refinement statistics (*Nature* template) is now included in the supplementary figures (SI Table 3, p.15).

(v) Please include a half map FSC and a model vs map FSC for the final reconstruction/model as well as an angular sampling plot in the figures. This is customary and helps readers to evaluate the quality of the reported reconstruction and the fit between model and map.

Response: FSC plot overlay and angular sampling plot is now included in the supplementary figures (**SI** Fig.9b, p.13).

(vi) The contour level provided by the authors in the pdb report appears off compared to the level used in the paper.

Response: The contour level in the PDB report is now closer to what is being used in the paper. The figures were prepared with a map sharpened using CryoEM tools in Coot – the contour level used for figures in the paper is 0.0288.

(vii) The authors must expand the the method section and provide more details, such as the exact ratios of receptor ligand, to name one incomplete information.

Response: Although the hormone:IR ratio was mentioned in the original manuscript the methods section (and main text as well) has been expanded to include more detailed information about complex formation, expression construct and AlphaFold prediction (marked in red in the main text).

REVIEWER #2

General Response: We are very grateful to the Reviewer for a very constructive critical, an detailed assessment of our manuscript. This was extremely helpful in getting the paper into fully professional format and content.

1. The manuscript has many typographic, spelling and grammatical errors and inconsistencies (suggesting to me that the manuscript may have been written in haste, perhaps in the face of competition). I recommend that the authors undertake a thorough proof-reading, as well as a check of the manuscript for self-consistency of style.

Response: We apologise for the number of technical deficiencies. The manuscript has been extensively edited and corrected.

2. In the Abstract:

(i) " dmIR binds three DILP5 molecules in an unseen arrangement": whereas what this means may become clear as the manuscript unfolds, it is not clear in the context of the abstract alone. It could mean that the authors don't observed the three DILP5 molecules at all! I suspect that the authors mean "hitherto-unseen" rather the "unseen"?

Response: This has been corrected along Reviewer's suggestion; in Abstract p.2: "hitherto-unseen" is used now.

(ii) "revealing unique structural signatures" is likely an over-statement. "Unique" with respect to what? The reported structural signatures may well occur across many insect species.

Response: Corrected: Abstract p.2: "showing also dmIR-specific features".

(iii) "provides structural proof of the evolutionary conservation..". Again, I would say this is an over-statement. I would be happier if it was worded "adds structural support to the evolutionary conservation...".

Response: We agree with the Reviewer, and this sentence has been replaced in the Abstract by:

“This work adds structural support to evolutionary conservation of the IIS axis at the IRs levels, underpinning also a better understanding of an important model organism.”

3. I may have missed it, but the authors make no reference to the N-terminal residues of dmIR that were excluded from their construct (which begins at residue 292).

Where precisely does the signal peptide lie? Even so, there remains a significant N-terminal "tail" prior to domain L1 - is there any indication of ordering of these residues within the map? Could they play a role in modulating either ligand binding or the structural conformations accessible to the receptor?

Response: The additional text has been added in the first part of the **Results**, p6:

“The dmIR-ECD construct used in this study is a codon optimised sequence derived from Uniprot sequence P09208. Residues 1-263 sequence, upstream of a predicted signal sequence (residues 264-290, see **Methods**), and downstream of the expected ectodomain (1-1309) were not included in the expression construct. A methionine was placed in front of the predicted

signal sequence with a C-terminal StrepII-tag and expressed in Sf9 cells using the baculovirus system.”

and in the **Methods**, p.17:

“The coding sequence beginning with the native signal sequence of dmIR (264H-NYSYSPGISLLLFILLANTLAIQAV-290V) was predicted using the SignalP 6.0 web service (64). DNA encoding the amino acids 264-1309 of dmIR was subcloned into the vector pBAC™4x-1 DNA (Novagen) downstream of the p10 promoter with the addition of a starting Methionine and C-terminal StrepII-tag.”

Asn335 is the first α -subunit residue observed in the map. Sentence on p.7 has been modified accordingly:

“The structure of the dmIR-ECD ($\alpha\beta$)₂ homodimer – with Asn335 as the first N-terminal residue observed in the maps - follows closely the multidomain order and size of hIR/hIGF-1R ectodomains (hIR-ECD/IGF-1R-ECD)”.

Whether the 291-334 part of the dmIR have any structural role – including the apo-dmIR-ECD - remains to be seen; the work on the apo-dmIR, and different constructs with variable length of the pre-L1 domain is under way (this is the first stage of our three years BBSRC grant).

However, it is worth to mention that the expression of the ‘full-length’ dmIR-ECD (1-1309): with the inclusion of residues 1-290 (with many repetitive sequences (e.g. (His)₄, (Gln)₄, (Arg)₄ etc.)) abolishes expression of this construct: both in our hands, and in the labs of NovoNordisk when the expression of the dmIR-ECD has been attempted as well. This supports further rather ambiguous character of 1-290 part of the Uniprot-reported dmIR sequence.

4. The reported cryoEM methodology and statistics need to be enhanced. The authors should

(i) include an FSC plot to convince the reader that the 3.9 Angs resolution has been correctly reported. In particular, I note some discrepancy between the resolution estimates contained in the PDB report and that stated in the manuscript.

Response: This plot has been provided in the composite **SI** Figure 9.

(ii) provide representative electron density at the residue level at selected locations within the structure in order to convince the reader that the resolution is as reported. Figures 2a and 3a do not suffice in this regard.

Response: Besides new figures with well-defined, sharpened maps (new Fig. 2, Fig. 3a), a new **SI** Figure 10 has been included with the examples of maps for the relevant part of the complex.

(iii) provide within the main figures a panel that indicates the putative (low-resolution) 3D conformation of the apo receptor extracted from the 3D classification (and indicate the associated class clearly within Figure S3).

Response: The putative apo 3D conformation of the dmIR-ECD is included in new Fig. 2a, and in the new **SI** Fig.9a (**SI** p.13), with a reference to the probable apo-form related class of this ectodomain.

(iv) whereas the PDB report contains information about model quality, the manuscript should also include at least some of this detail (e.g., Ramachandran statistics).

Response: New SI Table 3 (p.15 in the SI) – along Nature template for cryoEM data/model - contains this, and other relevant statistics.

5. The "extra" loop within the CR domain of dmIR is intriguing and, although not within the same location, is reminiscent of the additional loop within the CR domain of human EGFR that mediates dimerization of liganded EGFR. Is it possible for the authors to speculate on whether extra CR loop in dmIR could play a role in receptor-receptor interactions of either the liganded or ligand-free dmIR?

Response: This point has been addressed by modifying the text on p.8:

“The only significant deviation from the hIR fold is a large ~Gly491-Cys512 insert into the first part of the CR domain. However, the likely peripheral positioning of this loop - very disordered and not accountable in maps - suggests that it is not involved in a direct hormone binding, or dmIR-ECD dimerization, hence its role remains unclear. It cannot be excluded though that it is involved in a firmer attachment of the L1 domain to the dmIR protomers in the apo-dmIR, or, just the opposite, that it prevents a fully-down conformation of the L1-CR-L2 arm of the hormone-free protomer by clashing with the FnIII-3' domain (SI Fig. 4).”

Also, this loop has been modelled by AlphaFold for SI Fig. 4 (SI p.8), with a modified legend:

“SI Figure 4. Putative steric hinderance caused by dmIR-specific CR 491-512 insert.

The whole L1-CR domains of the dmIR-ECD were modelled here by AlphaFold (2), and the predicted (in green) and cryoEM structure L1 domains were superposed on their C α atoms. Subsequently, the dmIR L1 was superposed on the L1 domain from the fully-down, insulin-free lower-arm of the hIR (PDB ID: 6HN5+6HN4) one insulin site 1 complex (in white) (targets as in SI Table 1). This may suggest that the CR-insert in an unliganded dmIR protomer may cause a clash with FnIII-3' domain of the stem of the receptor (shown here in yellow, taken from the lower part of the hIR structure (PDB ID: 6HN4)). The red star indicates the overlapping regions.

Further analysis and comparison of the dmIR-ECD and EGFR loops would be very speculative and difficult due to a very different nature of dimerization of these receptors, and location of these loops in the context of different dimers of these receptors (i.e., 'outside' or 'inside' of a dimer).

6. At the beginning of the Results section, it is important for the authors to provide a brief summary of precisely what protein construct is being analysed, how it was expressed and purified and a statement that it was assessed as being "fit-for-purpose".

Response: Such summary has been provided at the beginning of the Results, p. 6:

“The dmIR-ECD construct used in this study is a codon optimised sequence derived from Uniprot sequence P09208. Residues 1-263 sequence, upstream of a predicted signal sequence (residues 264-290, see **Methods**), and downstream of the expected ectodomain (1-1309) were not included in the expression construct. A methionine was placed in front of the predicted

signal sequence with a C-terminal StrepII-tag and expressed in Sf9 cells using the baculovirus system.”

7. *There needs to be consistency in cited four-letter PDB IDs - the ms currently uses a mixture of upper and lower case.*

Response: All PDB IDs have been corrected and are in the upper case.

8. *SI Table 1. The inclusion of four decimal places in the r.m.s.d. values way exceeds their likely accuracy.*

Response: All rmsd values have been corrected to two decimal places.

9. *Figure 3c-f. The authors need to state what is being overlaid with what in generating the respective dmIR:hIR superpositions. Are the overlays based on the receptor domains alone (and if so which) or do they include the ligand(s)?*

Response: This point has been addressed in the legend of Figure 3 by inclusion of the sentence:

“**e – f** Binding of the DILP5 in site 1 down and site 1'; colour coding as above; numbering in grey italic refers to hIR (PDB ID: 6HN5). Superpositions in **c-d** has been done on the respective FnIII-1 domains, and in **e-f** on the L1/L1' domains without the ligands (see details of the superposition targets in the **SI Table 1**). Map in **3a** was sharpened by B-factor -50 Å².”

As the structures of the L1 domains are very similar/conserved in both receptors, they provide a very good superposition template, enabling easier assessment of specific features related to DILP5 binding in site 1 and site1'. This is also dictated by sequence differences in the sequence between dmIR/DILP5 and their human homologues, which can create additional 'bias' in comparative analysis. The same issues concern using the cores of FnIII-1 domains as the best targets in other superpositions, especially in analysis of site 2'.

10. *In the Figures and throughout, there are multiple versions of the prime symbol (') or ('). These should be standardized, my preference is for the former.*

All figures and the whole text has been modified in relevant places by ('), as requested by the Reviewer.

11. In Figures, the central beta sheet of domain L1 referred to as $\beta 2L1$, whereas in the text it is $L1\beta 2$.

Response: The use “ $L1\beta 2$ ” has been unified in all Figures and the main text, as requested, and along the established standard.

12. *The change in relative conformation of the domains in the receptor family upon ligand binding is likely correlates with the degree of association achieved between domains FnIII-3 and FnIII-3'. The authors report on p14 that the association between these domains is asymmetric, but no detail is given. Can the authors please clarify if these domains are in contact and, if so, via which elements, and does their association mimic any associations seen in liganded hIR and/or hIGF-1R? (note two typos in the associated para on p14 regarding PDB IDs: should be 4ZXB and 6HN4+6HN5).*

Response: The issue of association mentioned by the Reviewer has been addressed by adding new text on p.13.

“Impact of DILP5 binding on membrane-proximal domains.

The binding of three DILP5s to dmIR-ECD sites brings together the FnIII-3/3' domains, despite lack of any inter-protomers stabilising tags at their C-termini. The C-ends of FnIII-3/FnIII-3' domains are ~111 Å apart in the apo-hIR form (e.g. PDB ID: 4ZXB) to becoming ~16-20 Å close in some 1-3 insulin bound hIR conformers (e.g. PDB ID: 6HN4+6HN5, 7PG0-7PG4), and ~38 Å in hIGF-1R:hIGF-1 (PDB ID: 6PYH). In almost symmetrical T-shaped 4 insulins:IR complexes this separation is ~50 Å as the FnIII-3 C-termini (e.g., Lys917 in 6SOF) lie ‘outside’ of these domains that, nevertheless, are in a close contact. In the dmIR-ECD this distance (measured at Lys1310) is ~15 Å. However, the close contacts (~4-5 Å) between the β-sheet hairpins of the FnIII-3/3' domains (Glu1242-Cys1258 (Glu846-Cys869 in hIR)) are not as symmetrical as in some insulin:hIR complexes, where they interact through the centres of their edges (around His858), or their tips (as in 4 insulin:hIR, (PDB ID: 6SOF)). In dmIR-ECD, this interaction falls between Glu1252' and Lys1257, reflecting the upright shift (~6-10 Å) of the static protomer. In general, the overall arrangements of the FnIII-3/3' domains in DILP5:dmIR-ECD is close to the one observed in one-three insulin/hIR complexes (PDB ID: 6HN4+6HN5, 7PG0-7PG4). “

The PDB IDs typos has been corrected

13. *Fig 1d: are the sequence similarity numbers in the Figure percentages or number of conserved residues? This is not stated in the caption and is not clear.*

Response: These number correspond to sequence similarities in %; this has been clarified in Figure 1 legend.

14. *The authors need to include the missing reference in the caption to SI Figure 7.*

Response: Corrected – the reference has been added.

15. *On p17: " lacks any FnIII-1 domains joining – e.g. Leu-zipper-like tags ", the authors presumably mean "lacks any FnIII-3 domain joining tags"??*

Response: This has been corrected on p.16:

“any FnIII-3 domains association-inducing or stabilising tags.”.

16. *The using of AlphaFold. No literature reference is given in the ms or in the Reporting Summary to this software or to its version number. Given the relative newness of AlphaFold(1 or??),*

it would be opportune for the authors to comment on the perceived accuracy or otherwise of the AlphaFold prediction when assessed against the cryoEM maps.

Response: Both above points have been addressed by the new text in **Methods** p.19 and reference (70):

“Individual domains of dmIR-ECD were predicted using AlphaFold/2.0.0-foss-2020b (70) and docked as rigid bodies into the 4.0 Å resolution CryoEM maps using UCSF Chimera’s “Fit in map” function (71). The AlphaFold predictions were very accurate, with small rms deviations between the model and final structure domains (0.43, 0.68, 0.48, 0.53, 0.54 and 0.52 Å for L1, CR, L2, FnIII-1/2/3 respectively (C α atoms LSQ superposition in Coot (72)). The CR domain has been predicted with the lowest confidence.”

17. The authors may also discuss the propensity or otherwise for DILP oligomerization (i.e., with reference to that of human insulin), as it may well be that differences in association here with respect to the insulin:hIR system may be linked to this. In particular, DILPs lack the aromatic triplet that mediates human insulin dimerization.

Response: This has been addressed in the new text on p.12:

“Despite lack of human insulin-dimer stabilising triad (PheB24, PheB25, TyrB26 (SI **Figure 1**), dimerization of DILP5 was observed in more concentrated solution of this hormone (21). However, its crystal-reported dimer (PDB ID: 2WFV) is stabilised by the two - antiparallel β -sheet-forming - B-chains’ N-termini AspB1-BlyB7 β -strands (21) – not the B20-B28 chains. Such arrangements is not found in the dmIR-ECD:DILP5 complex, where the interface between site 1 and site 2’ hormones is limited, and their B1-B7 termini are in an approximately parallel – practically contact free - arrangement.”

18. p8, reference to SI Table 2 in 2nd para. I believe that this reference is correctly to SI Table 1.

Response: Corrected.

19. SI Fig. 8E. The ITC trace is marked as "typical" (which means more than one experiment was conducted). Why were these not all included to yield a mean K_d +/- SD? What type of analysis was conducted to reach the stated value of 274 nM (one-site or two-site, etc.)?

Response: We removed the word “typical” as other measurements were not suitable for statistical analysis. Data presented in this study was fitted using “One set of sites” binding model. You will notice the K_d has been amended to 498 nM. This reflects the re-analysis of the data using monomeric MW for DILP5. As result, $N = 2.7$ which is closer to what we see in the structure. This has been clarified in the legend of **SI Fig.8** and in the **Methods**. It may be argued whether monomer/dimer MW should be used in the context of ambiguous DILP5 dimerisation in solution. Ultimately, we decided here to use a monomer MW of the hormone, as it behaves as such on our SEC columns, and, importantly, it provides the N number of sites that reflect the structural observation. The IC_{50} competition-based assays are difficult here as the iodination of TyrA22 is very detrimental for DILP5 receptor binding.

20. SI Fig 7. There has been uncertainty regarding the pairing of the inter-alpha-chain disulfides at the Cys triplet in the insert domain of hIR. I note from SI Figure 2 that dmIR contains only two Cys residues at this point, suggesting a different structural motif at this site. Does the authors' map allow resolution of these bridge(s)?

Response: This has been addressed, in:

(i) Fig. 1d by changing the graphics (showing only two -SS- inter-ID domains bonds in dmIR), and adding a sentence in the figure legend:

“dmIR has only two inter-ID domains -SS- bonds as it does not have the equivalent of human Cys682.”

(ii) in the **Discussion** p.16, in the new sentence:

“However, the DILPs-induced allostery of the dmIR can diverge further from the human models as this receptor has only two of human three important – protomers-linking -cysteines in the ID segments (1010, 1012), with Tyr1009 replacing human third Cys682 (this region is not observed in the electron density maps).”

REVIEWERS' COMMENTS

Reviewer #1 (Remarks to the Author):

The authors have largely answered the majority of my questions in the revised version of the manuscript and supplied additional structural/structure related information. Overall, the manuscript's quality has greatly increased. The resolution and quality of the deposited model, however, remain sub-optimal, even with improved Ramachandran outliers at 0.28% or reduced clashes. However, I understand that obtaining a better data set requires enhancing protein quality and/or significantly increasing particle numbers. Furthermore, a higher "resolution" wouldn't significantly change the manuscript's claims. However, the authors should explicitly address these concerns, for example, in the relevant method section and/or figure legend, or in the discussion.

Major point:

(1) The authors now state a "supraphysiological" hormone:IR ratio of 5:1, which is insufficient. Please accept my apologies if my initial request was not clear enough. The DILP5:dmIR-ECD Kd of 498 nM found here is an order of magnitude larger than that reported for human insulin to human IR-ECD. As a result, it is critical to describe the exact molar concentrations of DILP5:dmIR-ECD for correct interpretation and for the sake of reproducibility. Could it be that, with such high Kd and ligand:receptor concentrations, the exhibited structure is not "in a previously unseen arrangement, displaying also dmIR-specific features"? The authors, in my opinion, ought to move the new paragraph at the beginning of the result section to the discussion section and expand it to emphasise this issue. The same applies to the word supraphysiological, which based on a 5:1 ratio cannot be judged/confirmed.

(2) The authors compare the herein disclosed structure to numerous recent cryo EM structures of human insulin. These structures, however, are difficult to directly compare because some are associated to the IR-ECD alone, while others are related to complete IR dimers in detergent micelles. Without modifying the exquisite visuals/figures, the authors must expand the discussion part to include ligand sensitivity and more. Otherwise, the assertion "in a hitherto-unseen arrangement" cannot be stated. The same holds true for the claims in "Impact of DILP5 binding on membrane-proximal domains".

(3) Because the authors compare different structures, I highly advise avoiding the term holo-dmIR-ECD and instead using dmIR-ECD. I understand why the authors used this term, however it is only intended to be used for full receptors, that is, receptors capable of transducing signaling. The herein reported word contains the receptor ectodomain only.

I'd like to congratulate the authors once more on their incredible success and highly recommend that the work be accepted for publication after the amendments are adopted.

No additional experiments are required for the additional amendments.

The authors have largely answered the majority of my questions in the revised version of the manuscript and supplied additional structural/structure related information. Overall, the manuscript's quality has greatly increased. The resolution and quality of the deposited model, however, remain sub-optimal, even with improved Ramachandran outliers at 0.28% or reduced clashes. However, I understand that obtaining a better data set requires enhancing protein quality and/or significantly increasing particle numbers. Furthermore, a higher "resolution" wouldn't significantly change the manuscript's claims. However, the authors should explicitly address these concerns, for example, in the relevant method section and/or figure legend, or in the discussion.

The 4 Å 'resolution' and reported quality of the model are well within the range of published IR cryoEM structures (3.1 – 7.6 Å). There is no need to enhance 'protein quality' as it is of high chemical purity – on par with other reported IRs, and a likely structural heterogeneity (its endogenous property) contributes to an overall 4 Å – not, let's say, 3 – 3.5 Å resolution. A higher dmIR-ECD *affinity* might be restored by the modifying the construct, e.g., by extending the FnIII-3 domains with some tags (Leu-zippers). However, we deliberately aimed to work with as native ECD as possible to obtain a snapshot of its conformational state: tags-unbiased structure hence with some possible physiological relevance. It's worth to mention that in >30 known IR structures only two of them (6SOF (4 Å) – one of the key quality and relevance benchmark in our work) and 7QID (5 Å), both by Guttman *at al.*, 2020) are 'native' (unmodified) ectodomains complexed with human insulins, and with membrane-proximal FnIII-3 domains in a close contact. Therefore, our approach paid off, providing a further and rare evidence about the interplay of hormone binding and allostery of the native IR-ECD.

Moreover, the map in the key hormone:dmIR-ECD interaction regions is unambiguous – corresponding there to ca 3.6 Å 'resolution' (SI Figure 9c), and any further enhancement of this map (or model quality) will not change the research outcomes of this work (as acknowledged by the Reviewer).

We are focusing now on the full-length dmIR and translation/validation of our ECD-based observations in transgenic flies.

We want to stress that we tried to avoid any unsupported claims. We underlined in several places the low-affinity nature of the dmIR-ECD, excess of hormone used during making of the complex, using frequently 'possible, putative, etc.' to underline – where appropriate - the ambiguity and scale of relevance of the interpretation.

Major point:

(1) The authors now state a "supraphysiological" hormone:IR ratio of 5:1, which is insufficient. Please accept my apologies if my initial request was not clear enough. The DILP5:dmIR-ECD Kd of 498 nM found here is an order of magnitude larger than that reported for human insulin to human IR-ECD. As a result, it is critical to describe the exact molar concentrations of DILP5:dmIR-ECD for correct interpretation and for the sake of reproducibility. Could it be that, with such high Kd and ligand:receptor concentrations, the exhibited structure is not "in a previously unseen arrangement, displaying also dmIR-specific features"? The authors, in my opinion, ought to move the new paragraph at the beginning of the result section to the discussion section

and expand it to emphasise this issue. The same applies to the word supraphysiological, which based on a 5:1 ratio cannot be judged/confirmed.

(i) Despite its lower 498 nM K_d of the dmIR-ECD, it does correspond 'structurally' to hormone coupling in an overall similar fashion as much higher affinity hIR-ECD constructs, e.g. SOF5, 7QID (~30 nM), or full length hIR, e.g. 7PG2, 7PG3, 7PG4 (~0.015 nM), while for some high-affinity hIR-ECDs (e.g. 6CEB, 6CE9) the stem of the IR is not defined in the maps. Therefore, "*Kd of 498 nM found here is an order of magnitude larger than that reported for human insulin to human IR-ECD*" is not detrimental to study dmIR-ECD. Whether it is a dmIR species-specific feature requires further research. (Please see also our response in (2) below).

(ii) We agree with the Reviewer about the term 'supraphysiological' and it has been removed, substituted in all relevant parts of the text by "five molar excess of the hormone over receptor."

(iii) The way of obtaining of the 5:1 hormone:IR ratio has been provided in detail in Methods, p.19:

"The dmIR-ECD:DILP5 complex was prepared by incubating equal volumes of 2.3 mg/mL (10 μ M) dmIR-ECD and 0.3 mg/mL (50 μ M) of DILP5 (gift by Novo Nordisk) in 50 mM Tris, 150 mM NaCl, pH 7.4 buffer, resulting in a five-molar hormone excess during making of this complex"

This should assure methodological reproducibility of the preparation of this complex.

(iv) The phrase "*in a previously unseen arrangement, displaying also dmIR-specific features*" has already been modified in the second version of the manuscript, as required by Reviewer 2, and replaced by his suggested "in a hitherto-unseen arrangement".

(v) We did not move the new paragraph at the beginning of the result section into discussion. It has also been requested there by Reviewer 2, and we agree with this addition in its current place as it provides the reader with a succinct characterization of the dmIR-ECD construct and DILP5 used in this study. Moreover, *Nature Communication* publication policy insists on much more concise and general character of the Discussion section, while Results covers more in-depth technical details as represented, for example, by this short introductory new paragraph here.

(2) *The authors compare the herein disclosed structure to numerous recent cryo EM structures of human insulin. These structures, however, are difficult to directly compare because some are associated to the IR-ECD alone, while others are related to complete IR dimers in detergent micelles. Without modifying the exquisite visuals/figures, the authors must expand the discussion part to include ligand sensitivity and more. Otherwise, the assertion "in a hitherto-unseen arrangement" cannot be stated. The same holds true for the claims in "Impact of DILP5 binding on membrane-proximal domains".*

We fully agree with the Reviewer about the complexity and challenges of inter-IRs comparisons, including our dmIR-ECD:DILP5 complex, and the lack of clear indication of the nature of the compared ECDs (i.e. bona fide ECD or part of the full-length IR) in the text; the PDB IDs alone are not sufficient in this matter. Therefore the figures concerning the ECD individual protomers and full ECD comparisons (Figure 5 and SI Figure 6) have been amended by clear graphical signage corresponding to the origin of the used/displayed ECD.

The already published insulin's:IR K_d s are in a broad range of ~60-0.8 nM, depending on the nature of the assay (e.g. PEG precipitation, hot-ligand competition assays with different protein/peptides immobilization approaches, microscale thermophoresis, ITC etc.), type of the IR construct (ECD, high-affinity restoring ECD-Leu-zipper, full length IR etc.), and the impact of the chemical micelle composition on the sensitivity of the IR has also been noted. Therefore we refrain here from a comparative use of these K_d s as it would likely result in a very convoluted, and frequently not fully compatible, arguments. A clear underlining of the low affinity nature of the dmIR-ECD (as it is presented) should provide sufficient context for a rather broad – and necessarily sketchy – tertiary and quaternary structures comparisons discussed here. Nevertheless, to highlight the point raised the Reviewer 1 we added in the Discussion, p.16:

“Despite a low ITC-measured K_D of 498 nM of the dmIR-ECD, the binding of three DILP5s to this constructs brings together the FnIII-3/3' domains which lack of any inter-protomers stabilising tags at their C-termini or any IR-intracellular components. Similar conformation of these domains has also been observed in higher affinity (~30 nM) tag-free hIR-ECD 4-insulins T- and 3-insulins-T-state complexes⁴¹ (PDB ID: 6SOF, 7QID). This supports further the correlation between hormone-coupling and convergence of IR-stem-protomers as an inherent feature of the ECDs in human and insect IR systems. “

The “a *hitherto-unseen arrangement*” has been requested by Reviewer 2, and, we think, this is not an overstatement or unsupported claim. Cealry, a complex with closely bound site 1/site2' ligands and site 1 present hormone has not been described before. The “*Impact of DILP5 binding on membrane-proximal domains*” has also not been questioned before by both Reviewers, and, we suggest, it can remain, especially in the context of the abovementioned addition to the text.

(3) Because the authors compare different structures, I highly advise avoiding the term holo-dmIR-ECD and instead using dmIR-ECD. I understand why the authors used this term, however it is only intended to be used for full receptors, that is, receptors capable of transducing signaling. The herein reported word contains the receptor ectodomain only.

We fully agree with the Reviewer 1 here hence all holo/apo terms used in the context of the dmIR-ECD have been removed, and substituted by “ligand-free” or “liganded” where applicable.